# Connexin Hemichannel Activation by S-Nitrosoglutathione Synergizes Strongly with Photodynamic Therapy Potentiating Anti-Tumor Bystander Killing

**DOI:** 10.3390/cancers13205062

**Published:** 2021-10-10

**Authors:** Chiara Nardin, Chiara Peres, Sabrina Putti, Tiziana Orsini, Claudia Colussi, Flavia Mazzarda, Marcello Raspa, Ferdinando Scavizzi, Anna Maria Salvatore, Francesco Chiani, Abraham Tettey-Matey, Yuanyuan Kuang, Guang Yang, Mauricio A. Retamal, Fabio Mammano

**Affiliations:** 1Institute of Biochemistry and Cell Biology (IBBC)-CNR, 00015 Rome, Italy; chiara.nardin@ibbc.cnr.it (C.N.); chiara.peres@ibbc.cnr.it (C.P.); sabrina.putti@cnr.it (S.P.); tiziana.orsini@cnr.it (T.O.); fmazzard@odu.edu (F.M.); marcello.raspa@cnr.it (M.R.); ferdinando.scavizzi@cnr.it (F.S.); annamaria.salvatore@cnr.it (A.M.S.); francesco.chiani@cnr.it (F.C.); abraham.matey@ibbc.cnr.it (A.T.-M.); 2Institute for Systems Analysis and Computer Science “A. Ruberti” (IASI)-CNR, 00168 Rome, Italy; claudia.colussi@cnr.it; 3Fondazione Policlinico Universitario Agostino Gemelli IRCCS, Catholic University of the Sacred Heart, 00168 Rome, Italy; 4Frank Reidy Research Center for Bioelectrics, Old Dominion University, Norfolk, VA 23508, USA; 5Shanghai Institute for Advanced Immunochemical Studies, ShanghaiTech University, Shanghai 201210, China; kuangyy@shanghaitech.edu.cn (Y.K.); yangguang@shanghaitech.edu.cn (G.Y.); 6Universidad del Desarrollo, Centro de Fisiología Celular e Integrativa, Facultad de Medicina Clínica Alemana, Santiago 7780272, Chile; mretamal@udd.cl; 7Department of Physics and Astronomy “G. Galilei”, University of Padova, 35131 Padova, Italy

**Keywords:** photosensitization, nitric oxide, purinergic signaling, calcium signaling

## Abstract

**Simple Summary:**

Bystander effects depend on direct cell-cell communication and/or paracrine signaling mediated by the release of soluble factors into the extracellular environment and may greatly influence therapy outcome. Although the limited data available suggest a role for intercellular gap junction channels, far less is known about the role of connexin hemichannels. Here, we investigated bystander effects induced by photodynamic therapy in syngeneic murine melanoma models in vivo. We determined that (i) photoactivation of a photosensitizer triggered calcium-dependent cell death pathways in both irradiated and bystander tumor cells; (ii) hemichannel activity and adenosine triphosphate release were key factors for the induction of bystander cell death; and (iii) bystander cell killing and antitumor response elicited by photodynamic therapy were greatly enhanced by combination treatment with S-nitrosoglutathione, which promoted hemichannel opening in these experimental conditions. Therefore, these findings in a preclinical model have important translational potential.

**Abstract:**

In this study, we used B16-F10 cells grown in the dorsal skinfold chamber (DSC) preparation that allowed us to gain optical access to the processes triggered by photodynamic therapy (PDT). Partial irradiation of a photosensitized melanoma triggered cell death in non-irradiated tumor cells. Multiphoton intravital microscopy with genetically encoded fluorescence indicators revealed that bystander cell death was mediated by paracrine signaling due to adenosine triphosphate (ATP) release from connexin (Cx) hemichannels (HCs). Intercellular calcium (Ca^2+^) waves propagated from irradiated to bystander cells promoting intracellular Ca^2+^ transfer from the endoplasmic reticulum (ER) to mitochondria and rapid activation of apoptotic pathways. Combination treatment with S-nitrosoglutathione (GSNO), an endogenous nitric oxide (NO) donor that biases HCs towards the open state, greatly potentiated anti-tumor bystander killing via enhanced Ca^2+^ signaling, leading to a significant reduction of post-irradiation tumor mass. Our results demonstrate that HCs can be exploited to dramatically increase cytotoxic bystander effects and reveal a previously unappreciated role for HCs in tumor eradication promoted by PDT.

## 1. Introduction

It has long been known that damage to cells directly hit by ionizing radiation can propagate to neighboring non-irradiated cells giving rise to the so-called radiation-induced bystander effect [1]. Bystander responses have been reported also downstream of other insults [2], including PDT [3,4,5,6,7], which requires the assimilation by tumor cells of a photosensitizer (PS) that remains inert and non-toxic until excited by light, whereas light-activated PS triggers cascades of highly cytotoxic reactive oxygen/nitrogen species (ROS/RNS) promoting tumor cell death [8].

Bystander effects depend on direct cell-cell communication and/or paracrine signaling mediated by the release of soluble factors into the extracellular environment and Cxs, a family of membrane proteins with four transmembrane helices, are involved in both type of processes [9]. Cxs form hexameric plasma membrane channels with large aqueous pores termed connexons or HCs [10]. Two HCs, each one from different cells in contact, may form an intercellular gap junction channel (IGJC) after docking head-to-head in the extracellular space and IGJCs have been implicated in bystander cell killing triggered by gene therapy for cancer [11], as well as ionizing radiation [12]. Although the limited data available also suggest a role for IGJCs in PDT-induced bystander effects, very little is known about the contribution of HCs [13].

In physiological conditions, HCs are usually closed, but a variety of factors may induce their opening, including protein phosphorylation or redox state, changes in pH or transmembrane voltage, metabolic inhibition, as well as NO and divalent cations [14]. In particular, the opening of several species of HCs is favored under conditions of low extracellular Ca^2+^ concentration ([Ca^2+^]_ex_) [14,15]. The open probability of HCs also varies in response to changes in the cytosolic Ca^2+^ concentration ([Ca^2+^]_cyt_), with a bell-shaped dependence peaking at 500 nM [16,17,18]. Open HCs can mediate the diffusive release of several paracrine messengers, most importantly ATP [19].

Our published results with aluminum phthalocyanine chloride (AlPcCl) in tumorigenic murine cell lines indicate that photoactivation of this PS triggers apoptosis in bystander tumor cells via propagation of intercellular Ca^2+^ waves that proceed from irradiated to bystander cells. AlPcCl activation also promotes NO production, a gaseous molecule that diffuses freely across plasma membranes and can enter bystander cells, in which NO concentration is further increased by a mechanism compatible with Ca^2+^-dependent enzymatic production [20,21,22].

Here, we analyzed bystander signaling and apoptosis induced by photoactivation of AlPcCl in vivo, in syngeneic mouse melanomas. We show that pharmacological treatments that favor HC opening enhanced ATP-dependent intercellular Ca^2+^ waves elicited by PS activation and increased bystander tumor cell killing, leading to a dramatic reduction of post-irradiation tumor volumes compared to PDT with PS alone.

## 2. Materials and Methods

### 2.1. Cells

#### 2.1.1. Lentiviral Vector Production

Lentiviral plasmid encoding GCaMP6s and mCherry were purchased from Addgene (Watertown, MA, USA; #80146, a gift from Dr. Darrell Kotton and pUltra-hot, and #24130, a gift from Dr. Malcolm Moore, respectively). Lentiviral plasmid encoding caspase-3 (Cas-3) indicator was engineered in the lab using the pUltra-hot lentiviral vector for bi-cistronic expression of mCherry and Cas-3 indicator (GANLS-DEVD-BNES, #50835, Addgene, a gift from Dr. Robert Campbell). Lentiviral plasmids encoding organelle-targeted Ca^2+^ indicators (R-CEPIA1er, G-CEPIA1er and CEPIA2mt) were a gift from Dr. Masamitsu Iino lab. Lentiviruses (LVs) were produced by transfection of HEK-293T cells (Cat. No. CRL-3216, American Type Culture Collection, ATCC, Manassas, VA, USA) using Lipofectamine 2000 (Cat. No. 11668-027, Thermo Fisher Scientific, Waltham, MA, USA). Supernatant containing viral particles was collected, cleared by centrifugation, and frozen at −80 °C for later usage.

#### 2.1.2. Engineering of B16-F10 Cells Expressing Fluorescent Indicators

B16-F10 mouse melanoma cells were obtained from ATCC (Cat. No. CRL-6475) and maintained in RPMI 1640-GlutaMAX (Cat. No. 61870-010, Thermo Fisher Scientific), supplemented with heat-inactivated fetal bovine serum (FBS, 10% *v*/*v*, Cat. No. 10270-106, Thermo Fisher Scientific) and penicillin/streptomycin solution (100 units/mL penicillin, 100 μg/mL streptomycin, Cat. No. 15070-063, Thermo Fisher Scientific). Cells were routinely tested for mycoplasma contamination by staining with 4′,6-Diamidine-2′-phenylindole dihydrochloride (DAPI, Cat. No. D1306, Thermo Fisher Scientific) followed by visual inspection at the fluorescence microscope. The day before LV infection, cells were plated at 30% confluence into a 35-mm Petri dish. Cells were pre-incubated in complete RPMI medium supplemented with hexadimethrine bromide (16 μg/mL, Cat. No. H9268, Merck KGaA, Darmstadt, Germany) at 37 °C for 30 min, then RPMI was replaced by 0.5 mL of LV-containing supernatant supplemented with hexadimethrine bromide (16 μg/mL). Cells were incubated for 3 h with the LV-containing medium alone to maximize infection efficiency, and then 1.5 mL of fresh culture medium was added to allow the incubation to continue overnight.

### 2.2. Animals

C57BL/6J mice were bred under controlled pathogen-free conditions at the National Research Council-Institute of Biochemistry and Cell Biology (CNR-IBBC), Infrafrontier/ESFRI-European Mouse Mutant Archive (EMMA), Monterotondo, Rome, Italy. At the end of experiments, mice were euthanized by cervical dislocation.

### 2.3. Surgical Procedures and Tumor Implantation

DSCs (Cat. No. SM100, APJ Trading Inc., Ventura, CA, USA) were mounted on 10–14-week-old male mice. During surgery, we removed a portion of the upper skin layer inside the DSC window using precision surgical scissors to expose the subcutaneous tissue. For multiphoton imaging experiments, the biopsy corresponded to the dorsal window area (12 mm ∅), whereas for longitudinal studies after PDT treatments, a punch biopsy was performed with a round scalpel (4 mm ∅, Cat. No. BP-40F, KAI Industries Co., Seki, Japan). The window chamber was then sealed with a sterilized, removable 12-mm ∅ round glass coverslip. On the same day, B16-F10 cells were infected with LV expressing the protein of interest. The day after, LV-transduced B16-F10 cells were collected in ice-cold HBSS (Cat. No. 14025-050, Thermo Fisher Scientific) for seeding onto the exposed subdermal tissue within the DSC biopsy. Chamber-bearing mice were transiently anesthetized with 2% gaseous isoflurane and a certain amount of cell suspension was dropped onto the subcutaneous tissue (5 × 10^5^ cells in a total volume of 30 μL for multiphoton imaging experiments and 3 × 10^5^ cells in a total volume of 8 μL for longitudinal studies). For longitudinal experiments after full PDT (see Section 2.8), mice were kept anesthetized for at least 40 min after cell seeding to promote sedimentation on subdermal tissue and increase uniformity of tumor growth across experiments. All mice were checked daily to monitor health conditions, melanoma progression and expression of fluorescent indicators.

### 2.4. Focal PDT (fPDT) and Multiphoton Intravital Microscopy of Melanomas Expressing Genetically Encoded Fluorescent Indicators

Four to five days after inoculation of indicator-expressing tumor cells, tumor-bearing mice were anesthetized with 2% isoflurane and maintained at 37 °C on a controlled heating pad. The 12-mm-round glass coverslip that sealed the DSC window was removed and the tumor grown in the exposed subcutaneous tissue was incubated for 1 h with PS (AlPcCl, 100 μM, Cat. No. 362530, Merck KGaA) dissolved in normal extracellular medium (NEM: 138 mM NaCl, 5 mM KCl, 2 mM CaCl_2_, 0.4 mM NaH_2_PO_4_, 6 mM D-Glucose, 10 mM HEPES, all purchased from Merck KGaA, pH 7.3), supplemented with the nonionic surfactant agent pluronic F-127 (1% *w*/*v*, Cat. No. 20053, AAT Bioquest, Sunnyvale, CA, USA) to facilitate PS solubilization. Next, mice were transferred to the multiphoton microscope stage, where the DSC was fixed to a custom-made holder to reduce motion artifacts due to animal breathing. The exposed tumor in the DSC was washed and incubated in fresh NEM as immersion medium for the objective for fPDT simultaneous to intravital multiphoton microscopy.

For fPDT, a single cell at the surface of the tumor was irradiated with light from a fiber-coupled 671-nm diode-pumped solid-state laser (Shanghai Dream Lasers Technology, Shanghai, China), as described in [22]. Briefly, the recollimated laser beam was injected into the microscope optical path just above the 60× water immersion objective (Nikon Fluor 60× Water, NA = 1.0, WD = 2 mm, Nikon, Tokyo, Japan) and focused into a 10-μm ∅ spot. During imaging experiments, the 671-nm (activation) laser was electronically triggered to deliver 5.5 ms flashes in the time interval between consecutive frames at an irradiance of ~5 × 10^6^ mW/cm^2^.

Multiphoton excitation of fluorescent indicators was provided by a femtosecond pulsed titanium-sapphire pump laser (Chameleon Ultra II Laser, Coherent Inc., Santa Clara, CA, USA) coupled to an optical parametric oscillator (MPX Chameleon Compact OPO, Coherent Inc.). Fluorescence signals were detected by cooled GaAsP photomultiplier modules (Cat. No. H7422-40, Hamamatsu Photonics, Hamamatsu, Japan) at 5.1 frames/s. Green fluorescent indicators (GCaMP6s, G-CEPIA1er, CEPIA2mt, and GANLS-DEVD-BNES) were excited at 920 nm and fluorescence emission was collected by a 525/40 nm band pass filter (Cat. No. FF02-525/40-25, Semrock, Rochester, NY, USA). R-CEPIA1er was excited at 1100 nm and mCherry was excited at 980 nm; red emission signals from these fluorescent indicators/proteins were selected by 612/69 nm band pass filter (Cat. No. FF01-612/69-25, Semrock).

For some fPDT experiments, GCaMP6s-expressing tumors in the DSC were incubated in Ca^2+^-free extracellular medium (CFEM, i.e., NEM devoid of Ca^2+^), supplemented with 5 mM of ethylene glycol-bis(β-aminoethyl ether)-N,N,N′,N′-tetraacetic acid (EGTA, Cat. No. E3889, Merck KGaA) with or without HC inhibitors, carbenoxolone (CBX, 100 μM, Cat. No. C4790, Merck KGaA), flufenamic acid (FFA, 100 μM, Cat. No. F9005, Merck KGaA), TAT-Gap19 (150 μM, peptide YGRKKRRQRRRKQIEIKKFK, Genosphere Biotechnologies, Clamart, France), and abEC1.1m (1 μM; for production methods see [23]; Shanghai Institute for Advanced Immunochemical Studies, ShanghaiTech University, 201210 Shanghai, China) for 20 min before the experiment. Tumors were randomly assigned to a single drug treatment and each one was used for several measurements performed in non-overlapping areas of the sample. Thereafter, washout was performed with fresh NEM for 30 min and experiments were repeated in CFEM supplemented with 5 mM of EGTA alone to confirm the reversible effect of HC blockers. We stopped data collection when at least *n* = 6 fPDT trials were performed in at least 2 tumors and pooled the results. Typical numbers of bystander cells (classified within a “bystander cell order” according to their distance from the irradiated cell, see Figure 1A) that were monitored in a single fPDT trial were: 1st order, 3; 2nd order, 6; 3rd order, 7; 4th order, 6, 5th order, 5; and 6th order, 2. The investigator was not blinded during administration of treatments or result assessment. No samples were excluded from analyses.

### 2.5. DAPI Uptake Assays

In vitro assays were performed on communication-incompetent (i.e., lacking ICGJ-mediated communication) HeLa DH cells (Cat. No. 96112022, Merck KGaA) cultured in DMEM (Cat. No. 41965039, Thermo Fisher Scientific) supplemented with 10% *v*/*v* heat-inactivated FBS and penicillin/streptomycin solution (100 units/mL). Cells were plated at 60% confluence onto glass coverslips and transfected after 24 h with plasmids encoding Cx43 or Cx26 fused to the yellow fluorescent protein Venus, using Lipofectamine 2000. Experiments were performed the following day. In vivo assays were performed on GCaMP6s- or mCherry-expressing tumors grown in DSCs for four to five days and prepared as described above.

For each sample, cells or tumors, multiple z-stacks (through-focus image sequences obtained with a 2 µm step size) were acquired with the multiphoton microscope before and after 30 min of incubation with DAPI (5 μM) and the HC opener/blocker under examination, dissolved in proper extracellular medium (as specified in figure legends). To test HC blockers, we used CFEM or CFEM supplemented with 5 mM of EGTA for in vitro and in vivo assays, respectively; to test GSNO (1 mM), we used NEM. All drugs were administered 20 min before starting the experiment. Multiphoton imaging was performed with the 60× Nikon objective (Figure 2B–D and Appendix A) or with a 25× water immersion objective (XLPlanN25XWMP2, NA = 1.05, WD = 2 mm, Olympus Corporation, Tokyo, Japan; Figure 5). For in vivo assays, we imaged cells within a depth of 20–60 µm from the tumor surface. DAPI was excited at 780 nm and imaged through a blue emission filter (460/50 nm, ET460/50m, Chroma Technology Corp, Bellow Falls, VT, USA).

### 2.6. Dual Focal Plane Multiphoton Imaging for Real-Time Detection of ATP Released by Tumor Cells In Vivo

Whole-cell biosensors for ATP detection (ATP-WCBs) were obtained by transduction of HEK-293T cells with a lentiviral vector (pHOT-P2Y2R, Addgene plasmid #159108) [24] encoding the P2Y_2_ G-protein-coupled receptor (R) that mobilizes Ca^2+^ from intracellular stores upon binding extracellular ATP with high affinity. ATP-WCBs were cultured in DMEM/F-12 (Cat. No. 11320074, Thermo Fisher Scientific) supplemented with 10% *v*/*v* heat-inactivated FBS and penicillin/streptomycin solution (100 units/mL). The day before the experiment, ATP-WCBs were plated at 50% confluence on 12-mm-round glass coverslips coated with laminin (2.5 μg/cm^2^, Cat. No. 11243217001, Merck KGaA). On the day of the experiment, ATP-WCBs were loaded for 30 min at 37 °C with fluorescent cytosolic Ca^2+^ dye Fluo-8H AM (5 μM, Cat. No. 21091, AAT Bioquest) dissolved in NEM supplemented with pluronic F-127 (0.1% *w*/*v*) and sulfinpyrazone (250 μM, Cat. No. S9509, Merck KGaA) to prevent dye secretion. For the detection of fPDT-induced ATP release by tumor cells in vivo, the coverslip plated with Fluo-8H-loaded ATP-WCBs was carefully flipped upside-down and placed within the frame of the skinfold chamber window, ~100–200 μm over the GCaMP6s-expressing tumor mass pre-loaded with AlPcCl, as described above. The narrow gap between tumor and tumor-facing ATP-WCBs was filled with CFEM to enhance HC opening, supplemented with ARL 67156 (100 µm, Cat. No. A265, Merck KGaA, an ecto-ATPase inhibitor), to avoid rapid degradation of the released ATP. To monitor fPDT-induced cytosolic Ca^2+^ wave in the tumor and concurrent ATP-WCB responses, we oscillated the 60× objective lens of the multiphoton microscope between the two focal planes using a piezo-electric actuator (PFM450E, Thorlabs, Inc., Newton, NJ, USA). For each plane, images were acquired at a rate of 0.5 frame/s. For these experiments, both Fluo-8H and GCaMP6s were excited at 920 nm and imaged through the 525/40 nm band pass filter.

### 2.7. Imaging Intracellular Responses to Exogenously Applied ATP

B16-F10 cells expressing genetically encoded fluorescent Ca^2+^ indicators targeted to the ER or mitochondria (G-CEPIA1er and CEPIA2mt, respectively) or the indicator for Cas-3 activity were plated at 30% confluence onto laminin-coated round glass coverslips 24 to 48 h before experiments. On the day of the experiment, cells were washed in NEM and placed on the stage of the two-photon microscope. For pressure application of exogenous ATP, glass microcapillaries (Cat. No. G85150T-4, Harvard Apparatus, Holliston, MA, USA) were pulled using a vertical puller (Cat. No. PP-830, Narishige Group, Tokyo, Japan) to a fine tip (4–5 µm ∅) and filled with CFEM supplemented by 100 nM of ATP. The filled pipette was connected with flexible tubing to a pneumatic Pico Pump (Cat. No. PV820, World Precision Instrument, Sarasota, FL, USA) and placed at a distance of 25–30 µm from the targeted cell with a micromanipulator (Cat. No. MX110, Siskiyou Corporation, Grants Pass, OR, USA). Fluorescent indicator signals were recorded at a rate of 4 frames/s before, during and after 125 s of ATP ejection from the microcapillary in CFEM (same excitation and emission wavelengths as in Section 2.4).

### 2.8. PDT Treatments and Longitudinal Studies by Time-Lapse Microscopy

For longitudinal studies, we performed PDT in two different configurations: spatially confined PDT (scPDT, in which a subarea of tumor surface was irradiated; see below and Figure 4A) and full PDT (in which the whole tumor surface was irradiated). To estimate the minimum sample size for each experimental group, we set a probability *α* = 5% for the type I error in the ANOVA test. Then, to fix *β* = 4*α* = 20% to obtain a test power of 1 − *β* = 80%, we computed the number of samples *n* using the formula:*n* = 2[(*z_α_*_/2_ + *z_β_*) × *σ*/Δ]^2^,(1)
with *z_α_*_/2_ = 1.96 and *z**_β_* = 0.8416. We assumed *σ* = 0.35 (35%) as estimation for variability of data (*σ*^2^ variance of data distribution) and Δ = 0.75 as an expected effect size (minimum normalized difference between averages that has biological significance). We treated in parallel pairs of mice (DSC surgery, B16-F10 cell inoculation, onset of treatment) and randomly assigned them to two different experimental groups. Experiments were repeated until a minimum number of samples was collected for each experimental group (*n* = 3 for scPDT, *n* = 4 for full PDT) and data were pooled. No mice were excluded from the study.

scPDT and full PDT treatments started two and four days after tumor cell seeding in the DSC punch biopsy, respectively. B16-F10 wild type cells were used for calcein-AM tumor viability assays, whereas B16-F10-mCherry were used for long-term longitudinal studies of PDT effectiveness, both in scPDT and full PDT experiments. On the day of the experiment, tumor-bearing mice were anesthetized with a cocktail of ketamine (90 mg/kg; Lobotor, Acme S.r.l., Cavriago, Italy) and medetomidine (0.5 mg/kg; Domitor, Orion Pharma, Espoo, Finland) via intraperitoneal injection. Next, tumors were sensitized by multiple injections of NEM supplemented with PS (100 µM, total microinjected volume = 10 μL) followed by incubation with PS-containing NEM for 1 h to ensure homogeneous PS absorption in the whole tumor mass. For calcein-AM viability assay, tumors were also loaded with NEM supplemented with calcein-AM (10 μM, total microinjected volume = 5 μL; Cat. No. C1430, Thermo Fisher Scientific). Thereafter, 10 μL of medium supplemented with the HC opener/blocker under examination were microinjected in the tumor. We used NEM (control); CFEM supplemented with EGTA (5 mM, for extracellular Ca^2+^ chelation); CFEM supplemented with EGTA and FFA (100 μM, for HC inhibition in EGTA conditions); NEM supplemented with GSNO (1 mM, for release of free NO, Cat. No. N4148, Merck KGaA); NEM supplemented with GSNO and FFA (100 µM, for HC inhibition in GSNO conditions); and NEM supplemented with GSNO and TAT-Gap19 (150 µM) or abEC1.1m (2 µM, for Cx43 or Cx26 HC inhibition in GSNO conditions, respectively). Microinjections were performed using a controlled micropump (PHD Ultra Nanomite, Cat. No. SKU 70-3601, Harvard Apparatus) that applied low and constant pressure to a 10-μL precision syringe (Cat. No. NANOFIL, World Precision Instrument) equipped with a 36-gauge beveled needle (Cat. No. NF36BV-2, World Precision Instrument). The syringe holder was mounted on a micromanipulator (Cat. No. MX110, Siskiyou Corporation) to carefully insert the needle in the tumor mass without damaging it.

Tumor surface irradiation for PS activation was provided by the 671-nm laser properly collimated in a 1.4-mm ∅ beam (scPDT) or a 5-mm ∅ beam (full PDT) by an adjustable aspheric collimator (Cat. No. CFC-2X-B, Thorlabs, Inc.). For scPDT, the DSC was covered with a round perforated matte mask (central hole diameter, 1.3 mm) to ensure confinement of PS activation. Laser irradiation parameters were:30 min of continuous irradiation, 5-mW average beam power across the mask hole, 378-mW/cm^2^ irradiance, 678-J/cm^2^ total delivered dose for scPDT;1 h of continuous irradiation, 48-mW average beam power, 245-mW/cm^2^ irradiance, 880-J/cm^2^ total delivered dose for full PDT.

Fluorescence images were acquired with a 2× super apochromatic microscope objective (TL2×-SAP, NA = 0.1, WD = 56.3 mm, Thorlabs, Inc.). Excitation light was provided by a fluorescence lamp (pE-4000, CoolLED Ltd., Andover, NY, USA) equipped with excitation filters suitable for calcein-AM (Cat. No. FF01-474/27-25, Semrock) and mCherry (Cat. No. FF01-565/24-25, Semrock), respectively. Fluorescence emission was collected by a 1.4-megapixel monochrome scientific CCD camera (1500M-GE, Thorlabs, Inc.) through a green (Cat. No. FF01-525/39-25, Semrock) or a red (Cat. No. FF01-615/45-25, Semrock) band pass emission filter.

### 2.9. Micro-Computed Tomography (µCT) Analysis of Tumor Volumes

Four days after full PDT treatments, mCherry-expressing tumors were carefully harvested under a fluorescence stereomicroscope (IC90 E, Leica, Wetzlar, Germany). Tumor masses were then fixed in 4% paraformaldehyde (PFA) for 6 h and dehydrated with consecutive washes in increasing concentration of ethanol up to 100% (50% overnight, 70% overnight, 96% 30 min, 100% overnight). The acquisition of tomographic datasets was performed in 0.5-mL plastic micro-tubes, through a high-resolution 3D µCT Imaging System (Skyscan 1172G, Bruker, Billerica, MA, USA), at 9.8 µm image pixel size. The reconstruction was performed using a built-in NRecon Skyscan Software (Version 1.6.6.0, Bruker). Manual image-by-image segmentations were performed to define specific volumes of interest for automated volume measurements, using the Bruker Micro-CT Analyser (Version 1.13) software.

### 2.10. Western Blot Analysis

Tumor masses were harvested five days after seeding either B16-F10 or B16-F10-GCaMP6s cells in DSCs. Protein extracts from either tumors or cultured cells were prepared in RIPA buffer (50 mM of Tris-HCl, pH 7.5, 150 mM of NaCl, 1 mM of EDTA, 1 mM of EGTA, 1% NP-40, 1% sodium deoxycholate) supplemented with 1 mM PMSF and protease inhibitor mix. Fifty micrograms of proteins were loaded on a 12% gel and western blot was performed according to standard procedures. Anti-Cx43 (1:8000, #ab11370, Polyclonal, Abcam, Cambridge, UK) and anti-Cx26 (1:250, #51-2800, Polyclonal, Thermo Fisher Scientific) primary antibodies were used to detect Cxs. Optical density signal was normalized to that of tubulin (1:2000, Cat. No. T6074, Monoclonal, Merck KGaA). Electrochemiluminescent signal derived from luminol-horseradish peroxidase-conjugated secondary antibody reaction was acquired and analyzed with an imaging system (Alliance, Uvitec Ltd., Cambridge, UK). Heart and kidney were used as positive control tissues.

### 2.11. Histology and Immunofluorescence

Entire tumor-bearing DSC tissues were collected and fixed in 4% PFA overnight for paraffin embedding. Microtome-sectioning was conducted to generate 8-μm sections that were deparaffinized and rehydrated. For histological analysis, standard hematoxylin and eosin staining was performed, and bright field images were acquired using an inverted microscope equipped with a 20× objective (NA = 0.8) and a color camera (THUNDER Imager DMi8, Leica). For immunostaining, antigen retrieval was performed in 1× citric buffer at 95 °C for 10 min, followed by permeabilization with 0.5% Tryton X-100 in TBS-T buffer for 10 min. After blocking for 1 h at room temperature in blocking buffer (10% FBS, 1% BSA in TBS-T), slides were incubated overnight in a humidified chamber at 4 °C with primary antibody anti-Cx43 (1:100, #71-0700, Polyclonal, Thermo Fisher Scientific), anti-Cx26 (1:200, #51-2800, Polyclonal, Thermo Fisher Scientific) or anti-P2Rs P2Y_1_R and P2Y_2_R (1:100, #APR-009 and #APR-010, respectively, Polyclonal, knock out-validated, Alomone Labs, Jerusalem, Israel). The next day, slides were washed and incubated with cross-adsorbed Alexa Fluor 488 secondary antibody (1:500, #A-11008, Thermo Fisher Scientific) for 1 h, then washed again and incubated with Alexa Fluor 594 conjugated anti-MelanA primary antibody A103 (1:100, Cat. No. sc-20032 AF594, Monoclonal, Santa Cruz Biotechnology, Inc., Dallas, TX, USA) for 2 h at room temperature. Afterwards, slides were washed and stained with DAPI (5 μM in TBS-T for 5 min). Washing was performed in TBS-T between all steps. Fluorescence images were acquired at confocal TCS SP5 microscope (Leica) equipped with a 63× oil immersion objective (HC PL Apo, UV optimized, NA 1.4, Leica). Alexa Fluor 488 fluorescence signal was excited by a 488-nm Argon laser and collected between 500 nm and 540 nm; Alexa Fluor 594 fluorescence signal was excited by a 543-nm HeNe laser and collected between 625 nm and 700 nm; DAPI fluorescence signal was excited by a 405-nm UV laser and collected between 415 nm and 480 nm. Using a sequential line-by-line protocol, images were acquired by averaging each line 16 times (total pixel dwell time, 2.5 ms; pixel size, 96 nm).

### 2.12. Image Quantification 

Image processing and data analysis were performed using Matlab (R2019a, The MathWorks, Inc., Natick, MA, USA) and the open-source software ImageJ/Fiji (ImageJ-win64). Extraction of fluorescence emission intensity *F* (fluorescent indicators, markers or dyes) and traces representing time-dependent relative variation of *F*,
Δ*F*/*F*_0_ = [*F*(*t*) − *F*_0_]/*F*_0_,(2)
where *F*_0_ denotes pre-stimulus value, were carried out as previously described [22].

Time integrals of Δ*F*/*F*_0_ traces were computed with the Matlab function trapz(). In time-lapse microscopy experiments (Figure 4, Figure 6 and Appendix A), tumor remission was estimated as decrease in fluorescent tumor surface area over time. The fluorescent area was defined as the portion of the image where pixel signal was higher than a fixed threshold previously set at image background level and measured with the ImageJ/Fiji plugin Measure. For DAPI uptake quantification (Figure 2, Figure 5 and Appendix A), nuclear fluorescence of a cell was measured within a round region of interest (ROI) as an average signal intensity of five consecutive z-stack frames that intercepted the nucleus of that cell, using the ImageJ/Fiji plugin Measure.

### 2.13. Statistical Analysis

Mean fluorescence traces are shown in the graphs as point-by-point mean ± standard error of the mean (s.e.m.) of Δ*F*/*F*_0_ or *F*/*F*_0_ signals. In histograms, mean values are quoted ± s.e.m. Box plots were generated with the Matlab function boxplot(). Sample sizes (*n*) for each experiment performed on the indicated number of tumors are declared in figure legends. Parametric analysis of variance was performed by one-way ANOVA or the two-tailed *t*-test (for multiple or pairwise comparisons, respectively); non-parametric statistical analysis (when normality of data was rejected by the Shapiro-Wilk test) was performed by the Kruskal-Wallis or the Mann-Whitney *U* test (for multiple or pairwise comparisons, respectively), as indicated in figure legends. In the case of multiple comparison testing, we used Bonferroni correction after the ANOVA test and Dunn-Sidak correction after the Kruskal-Wallis test; detailed results are provided in Appendix A. In all statistical analyses, a *p*-value (*p*) of < 0.05 indicates statistical significance: *, *p* < 0.05; **, *p* < 0.01; ***, *p* < 0.001; *p* > 0.05 is considered not significant (n.s.).

## 3. Results

### 3.1. Cx HCs Mediate the Propagation of Bystander Ca^2+^ Waves Elicited by Focal PS Activation In Vivo

For these experiments, we developed optically accessible B16-F10 murine melanomas in the DSC preparation [25] (see Materials and Methods and Appendix A). To reveal time-dependent variation of the [Ca^2+^]_cyt_, tumor cells were engineered to express the green fluorescent Ca^2+^ indicator GCaMP6s [26] prior to cell seeding in the DSC. We allowed tumors to develop in the DSC for 4 to 5 days, thereafter we focally stimulated a single cell in the PS-loaded tumor mass with laser light (671 nm) focused in a 10-μm ∅ spot, a procedure which we refer to as fPDT, and tracked the ensuing intercellular Ca^2+^ waves by intravital multiphoton microscopy [27] (Figure 1; see Materials and Methods for details). For clarity, melanoma cells within the field of view will be henceforth identified by a progressive “bystander cell order”, according to their distance from the irradiated cell (first neighbors of the irradiated cell are designated as 1st order cells and so on; Figure 1A, bottom right). To test HC involvement in Ca^2+^ wave propagation, we exploited the dependence of HC gating on [Ca^2+^]_ex_ [14,15] and compared experiments performed either in NEM (see Materials and Methods, standard control conditions) containing 2 mM of Ca^2+^ or after incubation in CFEM (see Ca^2+^-free conditions in the Materials and Methods) supplemented with 5 mM of EGTA. On average, lowering the [Ca^2+^]_ex_ in the DSC with EGTA increased ~four-fold the variation of [Ca^2+^]_cyt_ in bystander cells (Figure 1B,C). The simplest explanation for this seemingly paradoxical effect is that the ER was the main source of the observed [Ca^2+^]_cyt_ increments, which is consistent with previous results using AlPcCl as PS [22].

Prior work showed the B16-F10 mouse melanoma cell line expresses low basal levels of Cx26 and Cx43 [28,29]. To confirm the involvement of HCs in bystander signaling triggered by fPDT, we tested the propagation of Ca^2+^ waves in CFEM supplemented with 5 mM of EGTA, with or without the addition of non-specific Cx blockers CBX (100 µM) or FFA (100 µM) [30]. We also tested TAT-Gap19 (150 µM) and abEC1.1m (1 μM), i.e., two selective inhibitors of Cx HCs that do not affect IGJCs [23,31]. All four inhibitors decreased the amplitudes of bystander Ca^2+^ responses compared to CFEM with EGTA alone (Figure 2A and Appendix A). To corroborate the opening of HCs in low [Ca^2+^]_ex_ conditions, we used the dye uptake technique [32], both in vitro (Appendix A) and in vivo (Figure 2B–D). In addition, we assayed the expression of Cx43 and Cx26 by western blotting (Figure 2E) and immunofluorescence analyses (Figure 2F, we also tested the presence of Cx30 and Cx45, but they were not detected). Both Cx43 and Cx26 proteins were largely upregulated in B16-F10 melanomas grown in DSCs compared to B16-F10 cells cultured in vitro (Figure 2E). Consistent with these findings, CFEM supplemented with 5 mM of EGTA amplified fPDT-triggered intercellular Ca^2+^ waves substantially more in vivo (Figure 1) than in vitro [22]. Together, the results summarized above suggested that HCs play a critical role in bystander cell signaling associated with Ca^2+^ wave propagation in this syngeneic melanoma mouse model.

### 3.2. Extracellular ATP Is Required for fPDT-Induced Ca^2+^ Wave Propagation

Extracellular ATP may mediate intercellular Ca^2+^ waves by activating P2 purinoceptors on cell plasma membranes, with or without involvement of inositol 1,4,5-trisphosphate (IP_3_) diffusion through IGJCs [33]. Because HCs seem to be important for the intercellular Ca^2+^ waves described above, and given that open HCs are permeable to ATP, our next goal was to determine whether ATP was released during fPDT-triggered Ca^2+^ wave propagation in vivo. To test this hypothesis, we used ATP-WCBs developed in the lab, with ATP sensitivity in the nM range [24]. Cells were plated over the inner surface of the coverslip that sealed the DSC and loaded with a chemical fluorescent indicator selective for cytosolic Ca^2+^ (Fluo-8H, Figure 3A). The medium intercalated between tumor and ATP-WCBs in the sealed DSC was CFEM (to favor HC opening), supplemented with ARL 67156, an ecto-ATPase inhibitor [34], used to prevent rapid ATP degradation (EGTA was omitted because it interfered with ATP-WCBs activity).

Dual focal plane, intravital multiphoton confocal Ca^2+^ imaging showed that fPDT-evoked Ca^2+^ waves in GCaMP6s-expressing tumors were paralleled by time-delayed Ca^2+^ signals in overlying ATP-WCBs. Supplementing the extracellular medium with apyrase (APY, 250 U/mL), an enzyme that catalyzes ATP hydrolysis [35], reversibly inhibited Ca^2+^ wave propagation in both tumor and ATP-WCBs (Figure 3B,C). In the absence of a PS-loaded tumor, exposure of ATP-WCBs to the laser beam used for fPDT did not elicit any Ca^2+^ response (Appendix A). We obtained similar results from fPDT trials performed in the same conditions as those presented in Figure 1 and Figure 2A (incubation of DSC tumor with CFEM supplemented with 5 mM of EGTA, Figure 3D). Together, these in vivo fPDT results indicate that (i) ATP-WCBs responded to the ATP released by tumor cells after PS activation; and (ii) ATP release was required for Ca^2+^ wave propagation in bystander tumor cells.

When extracellular Ca^2+^ is chelated with EGTA, the P2 purinoreceptors that can still generate Ca^2+^ signals in response to extracellular ATP, by mobilizing Ca^2+^ from the ER through IP_3_Rs, are the G-protein-coupled P2YRs [36]. To confirm their expression in our tumor samples, we performed immunofluorescence experiments with specific and knock-out-validated antibodies against P2Y_1_R and P2Y_2_R and determined that both were detectable (Appendix A).

### 3.3. Intratumor Injection of EGTA Prior to PS Activation Boosts Bystander Cell Killing In Vivo

Based on the findings reported above, we hypothesized that potentiating HC activity by lowering the [Ca^2+^]_ex_ in the tumor microenvironment with EGTA might broaden the extent of cytotoxic bystander effects in this melanoma mouse model. To test this hypothesis, we seeded B16-F10 melanoma cells on the dermis exposed by performing a 4-mm ∅ punch biopsy within the DSC. Two days after seeding, we loaded the melanoma with both PS and a live cell marker (calcein-AM) that becomes fluorescent in (and therefore labels only) cells with active intracellular esterases [37]. Twenty minutes prior to PS activation, CFEM supplemented with 5 mM of EGTA or NEM (control conditions) was administered via intratumor microinjection. Next, we covered the DSC with a perforated matte mask and limited laser irradiation to within 10% of the melanoma area (Figure 4A), a procedure we refer to as scPDT (see Materials and Methods). Thereafter, we removed the mask and performed time-lapse intravital fluorescence microscopy with a 2× objective to estimate post-irradiation bystander cell viability in a field of view encompassing the whole tumor surface. Finally, at the end of a three-hour observation period, we reloaded the melanoma with calcein-AM in the DSC to counteract the unavoidable slow leakage of dye through the intact cell membrane of live cells [38]. In control conditions, ~50% of the melanoma surface remained fluorescent after calcein-AM reloading (Figure 4B,C, top). In contrast, in samples treated with CFEM supplemented with 5 mM of EGTA, we observed a near-complete loss of post-irradiation fluorescence, which persisted after calcein-AM reloading (Figure 4B,C, bottom). Neither laser irradiation in the absence of the PS nor EGTA microinjection without laser irradiation significantly affected calcein-AM fluorescence emission (Appendix A), indicating that its post-irradiation waning was genuinely caused by PS activation. A possible explanation for the persistent loss of calcein-AM fluorescence after dye reloading is that PS activation induced a massive HC opening, which would doom cells exposed to prolonged membrane permeabilization via HCs [14]. Consistent with this interpretation, chelation of extracellular Ca^2+^ with EGTA, a condition that promoted HC opening in our melanoma model (Figure 2B–D), enhanced bystander cell killing following PS excitation.

Since mobilization of internal Ca^2+^ stores seemed to be a key process in melanoma response to PS activation in vivo, and given that ER-mitochondria Ca^2+^ transfer is a key regulatory mechanism in Ca^2+^-dependent apoptosis and anti-cancer therapies [39,40,41,42], we investigated inter-organellar Ca^2+^ signaling in B16-F10 tumors grown in DSCs and expressing Ca^2+^-selective genetically encoded indicators targeted to subcellular compartments (Figure 4D) [43]. Following focal PS activation in NEM, these indicators revealed a rapid release of Ca^2+^ from the ER (Figure 4E,F, top) paralleled by Ca^2+^ uptake into mitochondria (Figure 4E,F, bottom) in both irradiated and bystander cells. Additional experiments on B16-F10 melanomas expressing a fluorescent indicator for Cas-3 activity (Figure 4G,H) [44] revealed Cas-3 cleavage in bystander cells within tens of seconds of laser irradiation (Figure 4I,J and Appendix A). Together with prior work [20,21,22], these results indicate that PS activation efficiently triggered Ca^2+^-dependent apoptotic pathways [39,40,41,42] in both irradiated and bystander melanoma cells.

To examine the consequences of these phenomena, we performed a longitudinal study based on time-lapse microscopy up to three days after scPDT in B16-F10 melanomas tagged with mCherry, a cytosolic fluorescent marker (Appendix A). mCherry fluorescence persisted in more than 75 % of the tumor surface at 24 h post-irradiation, slowly decreasing to ~65 % within 3 days in samples microinjected with NEM (control conditions) prior to laser irradiation. In contrast, mCherry fluorescence become undetectable within the first 24 h post-irradiation in samples microinjected with CFEM supplemented with 5 mM of EGTA, indicating that melanoma cells were almost completely eradicated from the tumor surface (consistent with the results of Figure 4B,C). The addition of FFA (100 μM) to the microinjected CFEM supplemented with 5 mM of EGTA reduced bystander cell killing to the level of controls performed in NEM (Appendix A).

### 3.4. GSNO Activates HCs and Improves PDT Outcome in Murine Melanoma In Vivo

Together, the results summarized above indicate that intratumor injection of EGTA increased HC-mediated bystander tumor cell death induced by scPDT. However, the beneficial impact of chelation therapy has been questioned [45,46]. An alternative procedure to bias HCs towards the open state is the use of GSNO, an endogenous NO donor [47] that acts in a metabolic state-dependent manner [48,49,50].

To determine whether the antitumor response induced by PDT could be enhanced by GSNO, we used mCherry-expressing melanomas grown for 4 days in the 4-mm ∅ punch biopsy. We first microinjected tumors with GSNO (1 mM) dissolved in NEM 20 min before performing in vivo DAPI uptake assays. Multiple through-focus image sequences (z-stacks) were acquired before and after 30 min of incubation with DAPI (5 µM). GSNO significantly enhanced DAPI uptake in vivo compared to controls (microinjection of NEM) and the effect of GSNO was inhibited by FFA (Figure 5), suggesting that GSNO promoted the opening of HCs in our syngeneic melanoma model. On this ground, we irradiated the whole surface area of other PS-loaded melanomas with a 5-mm ∅ laser beam for 1 h, a procedure hereafter referred to as full PDT (see Materials and Methods) in the absence or in the presence of microinjected GSNO (1 mM). Thereafter, we monitored post-irradiation mCherry fluorescence signal loss by time-lapse microscopy (Figure 6A,B). Four days after treatments, animals were humanely euthanized, tumors were harvested, and their volumes were measured with high precision by μCT analysis (see Materials and Methods) [51] (Figure 6C,D). A parallel set of samples was used for histology (Figure 6E). On average, full PDT reduced tumor volumes in samples microinjected with GSNO to 14% of the value achieved without GSNO (*p* = 0.044). Importantly, addition to the microinjected GSNO of specific HC blockers, either TAT-Gap19 (150 µM) or abEC1.1m (2 µM), abolished the observed reduction of tumor volumes, which were comparable to those obtained in control conditions (*p* = 0.80 for TAT-Gap19, *p* = 0.99 for abEC1.1m). In contrast, the volumes of GSNO-treated vs. NEM-treated melanomas that had not been laser irradiated after PS loading were not significantly different (*p* = 1). Consistent with these results, in B16-F10 cells stimulated in vitro with exogenously applied ATP (100 nM in CFEM), we detected Ca^2+^ release from the ER (via G-protein-coupled P2YRs, Appendix A) and Ca^2+^ uptake into mitochondria, but we also detected Cas-3 activation only in cell cultures that had been pre-incubated with 100 µM of GSNO (Appendix A).

Together, these experiments highlight a central role for HC activation in PDT for cancer and suggest that GSNO is a candidate adjuvant for PDT in clinical settings.

## 4. Discussion

We conducted this study in a syngeneic mouse melanoma model, working with B16-F10 cells grown in the DSC preparation [25] that allowed us to gain optical access to the processes triggered by PDT. By using a combination of intravital multiphoton microscopy [27], ATP-WCBs [24], and genetically encoded fluorescent indicators [26,43,44] expressed in B16-F10 tumors, we determined that (i) photoactivation of a PS suitable for melanoma treatment efficiently triggered Ca^2+^-dependent cell death pathways in both irradiated and bystander melanoma cells; (ii) HC activity and ATP release were key factors for the induction of bystander cell death; and (iii) bystander cell killing and antitumor response elicited by PDT were greatly enhanced by combination treatment with GSNO, which promoted HC opening in these experimental conditions.

The scheme that emerges from our present work and published results is one in which both NO and Ca^2+^ signaling, mediated chiefly by the ATP/P2YR/PLC/IP_3_ signal transduction cascade, are essential constituents of propagation and regeneration of ROS and NO production in the bystander cells, and are ultimately responsible for bystander effects.

### 4.1. HC-Mediated Ca^2+^ Signaling and ATP Release as Critical Components of Bystander Tumor Cell Killing

Bystander effects triggered by PS activation were mediated by intercellular Ca^2+^ waves that required release of ATP from cell cytosol to extracellular milieu through HCs. Both Cx43 and Cx26 were largely upregulated in B16-F10 cells grown in vivo (this work) compared to cells cultured in vitro **[28,29]**. Consistent with these findings, lowering the [Ca^2+^]_ex_ with EGTA, which removes HC blockade due to extracellular Ca^2+^ [14,15], amplified intercellular Ca^2+^ waves triggered by focal PS activation substantially more in vivo (this work) than in vitro [22], supporting a role for HCs in melanoma sensitization to bystander cell death.

In the extracellular milieu, the ATP released through HCs may couple with Ca^2+^-mobilizing purinoceptors on the plasma membrane of neighboring cells and develop intercellular Ca^2+^ waves [24,33,38,52]. Since the open probability of HCs is modulated by the [Ca^2+^]_cyt_ [16,17,18], a positive-feedback loop involving the control by [Ca^2+^]_cyt_ of HCs may lead to an overall augmented ATP-induced ATP release [52] that would increase both the volume invaded by Ca^2+^ waves and the [Ca^2+^]_cyt_ increments in individual bystander cells, as reported here (Figure 1).

To inhibit Ca^2+^ wave propagation in vivo, we used CBX, FFA, TAT-Gap19, and abEC1.1m. The first two drugs are widely used non-specific inhibitors of connexin-made channels with well-known side effects [53,54,55]. In contrast, TAT-Gap19 and abEC1.1m are among the most selective HC inhibitors currently available [23,31]. We showed here that FFA, as well as TAT-Gap19 and abEC1.1m, not only hampered Ca^2+^ wave propagation but also the uptake of DAPI into melanoma cells (Figure 2A–D). Dye uptake (with DAPI and other fluorescent permeant molecules) is a standard method for studying HC permeability and dynamics in a quantitative, sensitive, and versatile manner [32]. Both chelation of extracellular Ca^2+^ with EGTA and intratumor injection of GSNO, which opens HCs even when [Ca^2+^]_ex_ is at mM levels [48,49,50], increased DAPI uptake in vivo. When used in combination with PS activation, both EGTA and GSNO increased bystander cell killing, and GSNO dramatically reduced the residual melanoma mass after PDT (Appendix A and Figure 6). Although GSNO can affect several targets, the fact that its effects were drastically inhibited by FFA, TAT-Gap19, and abEC1.1 support our hypothesis that HCs are critical components of bystander signaling in anti-melanoma PDT. Considering the ability of PDT to promote immunogenic cell death (ICD) mediated by the release of damage-associated molecular patterns (DAMPs), and given that ATP is a prime DAMPs [56], it is tempting to speculate that ICD contributed to the remarkable reduction of post-PDT tumor mass in the presence of GSNO. Further studies are required to address this crucial issue.

The rise in [Ca^2+^]_cyt_ triggered by PS activation was dominated by Ca^2+^ release from the ER. IP_3_R channels serve as the principal mechanism to mobilize Ca^2+^ from ER stores [57]. Our prior work with B16-F10 melanoma cells in vitro [22] suggested that part of the [Ca^2+^]_cyt_ increase was mediated by the superoxide anion produced by PS activation and its breakdown product hydrogen peroxide (H_2_O_2_), which can directly activate IP_3_Rs by oxidation of thiol groups of critical cysteine residues [58]. In addition, increased levels of H_2_O_2_ have been shown to promote HC opening independently of [Ca^2+^]_ex_ in cultured cells in vitro [59]. We also previously highlighted the importance of ER-mitochondria Ca^2+^ transfer in the induction of a second generation of mitochondrial ROS both in irradiated and bystander cells following PS activation in vitro [22]. Here, in vivo experiments with ATP-WCBs and genetically encoded fluorescent indicators confirmed ATP-dependent Ca^2+^ efflux from the ER, paired with mitochondrial Ca^2+^ uptake followed by rapid activation of Cas-3 both in irradiated and bystander cells (Figure 4). In B16-F10 cultured cells, activation of P2YR by extracellular ATP was not sufficient, per se, for Cas-3 activation, but required the presence of the NO donor GSNO (Appendix A). On this ground, a schematic model for HC-related signaling downstream of PS photoactivation is presented in Figure 7 based on current understanding of mitochondria-associated ER membrane structure and function [60]. The model summarizes graphically the most likely processes that underlie the catastrophic combination of ER-mitochondrial Ca^2+^ signaling events, which are crucial to regulate tumor cell fate by promoting bystander cell death. Remarkably, (i) these molecular processes occurred in non-irradiated/bystander tumor cells within tens of seconds, with a time delay dependent on the distance from the irradiated cell; and (ii) bystander killing was augmented under conditions that favored HC opening.

### 4.2. Translational Significance of These Studies

Wu and colleagues showed that Cx-made channels increase Photofrin-mediated PDT phototoxicity in tumors derived from Cx-transfected HeLa cells and suggested, as a beneficial therapeutic strategy, a transitory elevation in Cxs expression [71,72]. However, Cxs play complex roles in tumor progression [73,74], therefore controlling their intratumor expression can be challenging; even more so given that Cxs have been found up-regulated in, and correlated to, increased metastasis [28,29]. In contrast, intratumor administration of a transiently active HC opener, such as GSNO [48,49,50], as carried out in this study, appears more feasible and far less risky, also considering that GSNO is an endogenous NO donor [47] and that NO is a byproduct of AlPcCl photoactivation [20].

In published work, NO improved lopinavir anti-cancer activity in syngeneic B16 models [75] and was required to efficiently induce H_2_O_2_-mediated cytotoxicity in B16 cultured cells [76]. Rapozzi and colleagues proposed a synergistic effect of NO donors and PS to amplify PDT-mediated cytotoxicity [77,78]. However, accumulating evidence reveals that, in sub-optimal PDT treatments, low-level photodynamic stress can induce NO-mediated tumor resistance and pro-survival effects in different cancer types [79,80]. Since the overall amount of produced NO depends on the PS used, irradiation settings, GSNO concentration, and targeted tissue, the parameters for combined treatment must be carefully determined in clinical settings. Development of novel selective HC-activating antibodies to be used as PDT adjuvants, instead of aspecific compounds, would be an important step towards future clinical applications. Further improvements in PDT are likely to accrue also from PS with large ROS/RNS generation capacity and absorption bands in the near infrared region [81,82], where tissue absorption is minimal [83].

## 5. Conclusions

Our study in a clinically relevant tumor model demonstrates that bystander effects in PDT proceed by conveying death signals from irradiated to bystander tumor cells via ATP-dependent ATP release through HCs, and these signaling mechanisms can by amplified by HC openers. These findings have a clear translational significance and suggest that modulators of HC activity can work synergistically with PDT to improve tumor eradication.

Recently, Ramadan and colleagues described a role for HCs in radiation-induced damage in endothelial cells [84], and ATP release through HCs following γ-radiation has been reported in cultured B16 melanoma cells [85], suggesting that the strengthening of ATP-dependent paracrine signaling demonstrated in this study could also be beneficial in other therapeutical modalities, such as radiotherapy, which is known to evoke bystander effects.

## Figures and Tables

**Figure 1 cancers-13-05062-f001:**
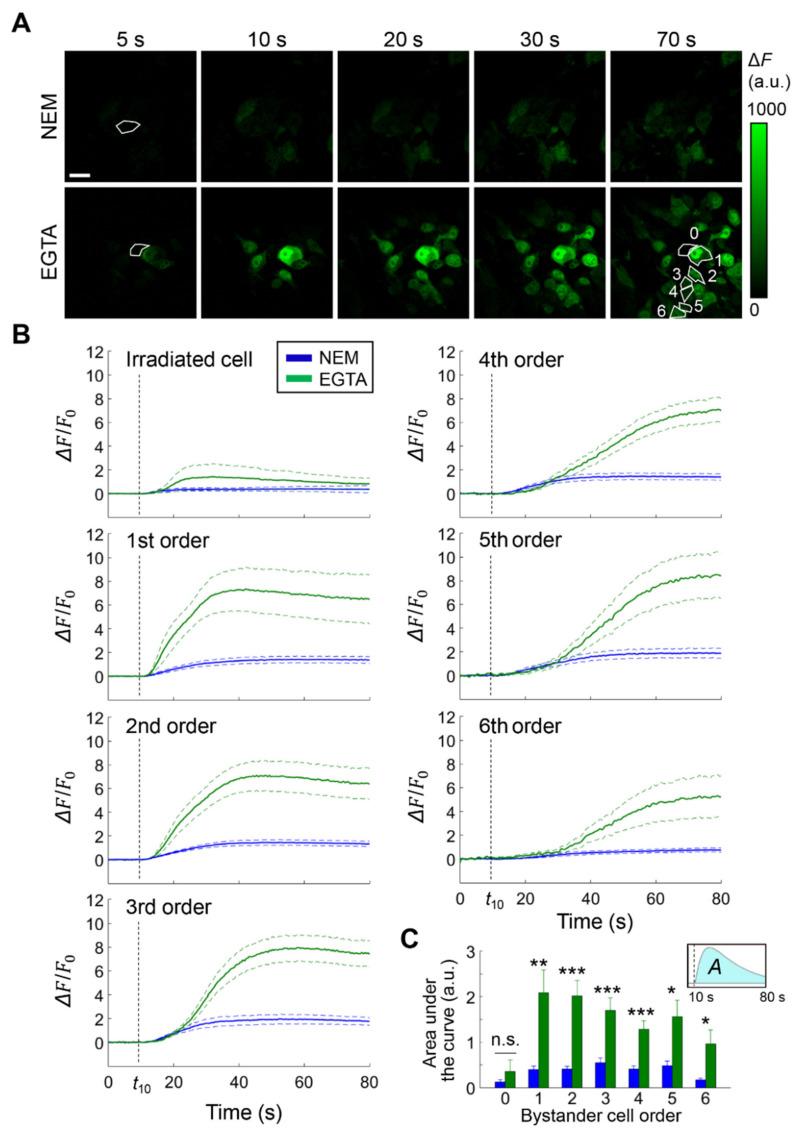
Intercellular calcium (Ca^2+^) waves triggered by focal photodynamic therapy in vivo. (**A**) Shown are GCaMP6s fluorescence emission (*F*) variations (Δ*F* = *F − F*_0_, where *F*_0_ = pre-stimulus value) at different time points after the onset of laser irradiation in standard conditions (normal extracellular medium containing 2 mM of Ca^2+^, NEM) or after 20 min incubation in Ca^2+^-free extracellular medium supplemented with ethylene glycol-bis(β-aminoethyl ether)-N,N,N′,N′-tetraacetic acid (EGTA, 5 mM). The contour of the irradiated cell is highlighted in the images captured at 5 s. Bystander cells were identified by ordinal numbers according to the distance from the irradiated cell (see, for an example, the contoured cells in the image at 70 s in EGTA conditions); scale bar: 20 µm; (**B**) Single-cell Δ*F*/*F*_0_ traces [mean (solid lines) ± standard error of the mean (s.e.m., dashed lines)] generated as pixel signal average within regions of interest contouring the cell focal plane section for each bystander cell order (from 1st to 6th); pooled data from *n* ≥ 6 experiments in 3 tumors for both conditions: green traces, EGTA; blue traces, NEM; vertical dashed lines mark the onset of irradiation at *t* = 10 s; (**C**) Area under Δ*F*/*F*_0_ curves (*A*, inset) computed between *t* = 10 s and *t* = 80 s (mean ± s.e.m.) vs. bystander cell order (abscissa): green bars, EGTA; blue bars, NEM; a.u., arbitrary units; n.s., not significant; *, *p* < 0.05; **, *p* < 0.01; ***; *p* < 0.001; the Mann-Whitney *U* test.

**Figure 2 cancers-13-05062-f002:**
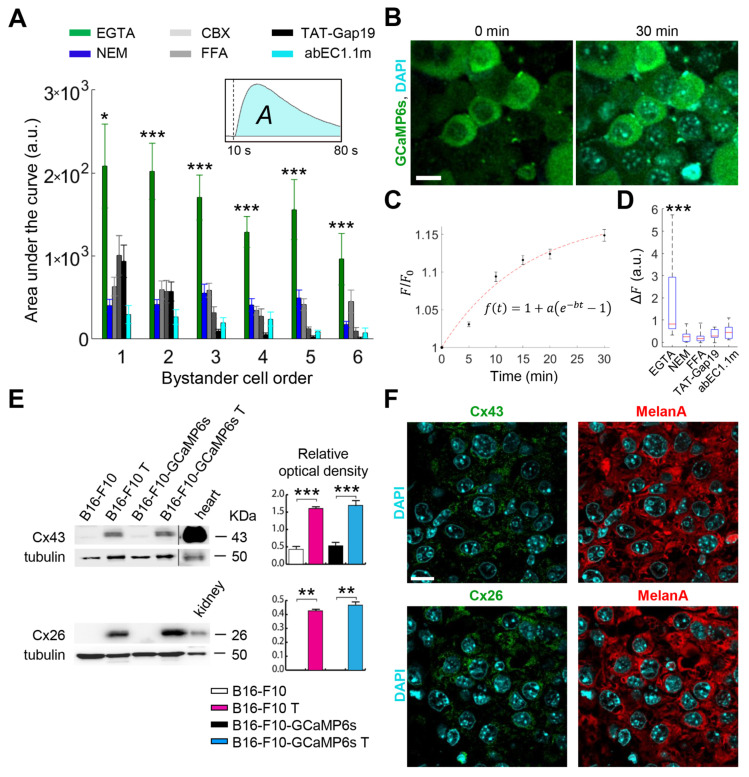
Connexin (Cx) hemichannels (HCs) expressed in melanoma cells mediate the propagation of calcium (Ca^2+^) waves induced by focal photodynamic therapy (fPDT) in vivo. (**A**) Pooled results of fPDT trials in GCaMP6s-expressing dorsal skinfold chamber (DSC) tumors in the following conditions: Ca^2+^-free extracellular medium (CFEM) supplemented with ethylene glycol-bis(β-aminoethyl ether)-N,N,N′,N′-tetraacetic acid (EGTA, 5 mM); normal extracellular medium containing 2 mM of Ca^2+^ (NEM, control); CFEM supplemented with EGTA (5 mM) plus carbenoxolone (CBX, 100 µM) or flufenamic acid (FFA, 100 µM) or TAT-Gap19 (150 µM) or abEC1.1m (1 μM). The histogram shows the area under GCaMP6s Δ*F*/*F*_0_ traces (*A*, inset) computed between the onset of fPDT (*t* = 10 s) and the end of the observation time window (*t* = 80 s) for each bystander cell order (abscissa); pooled data [mean ± standard error of the mean (s.e.m.)] for *n* ≥ 6 experiments in at least 2 different tumors for each condition. a.u., arbitrary units; *, *p* < 0.05; ***, *p* < 0.001, the Kruskal-Wallis test (for post hoc pairwise comparisons, see Appendix A). Data for EGTA and NEM conditions are also shown in Figure 1C; (**B**–**D**) In vivo 4′,6-Diamidine-2′-phenylindole dihydrochloride (DAPI) uptake experiments: GCaMP6s-expressing DSC melanomas were incubated with DAPI (5 μM) dissolved in: CFEM supplemented with 5 mM of EGTA; NEM; CFEM supplemented with 5 mM of EGTA plus FFA (100 µM) or TAT-Gap19 (150 µM) or abEC1.1m (1 μM). Fluorescence images were acquired at 5 min intervals up to 30 min; (**B**) Representative images acquired before (*t* = 0 min) and after 30 min of DAPI incubation in EGTA conditions; scale bar: 20 µm; (**C**) Relative variation of nuclear DAPI fluorescence intensity [*F*(*t*)/*F*_0_] in tumor cells vs. time during dye uptake in EGTA conditions (mean ± s.e.m., *n* = 20 cells, 2 tumors). The red dashed line was computed by data fitting with the shown function *f*(*t*) (parameter values: a = −0.1754, b = 0.0647 min^−1^); (**D**) Box plots showing the distributions of Δ*F* = *F*(30 min) − *F*_0_ for DAPI measured in *n* ≥ 12 nuclei for each condition. Red horizontal bars indicate the median. ***, *p* < 0.001, the Kruskal-Wallis test (for post hoc pairwise comparisons, see Appendix A); (**E**) Representative western blots for Cx43 (top) and Cx26 (bottom) expression in tumors (T) derived from B16-F10 or B16-F10-GCaMP6s cells and grown in DSCs (denoted as B16-F10 T and B16-F10-GCaMP6s T, respectively) compared with B16-F10 or B16-F10-GCaMP6s cells grown in culture dishes; graphs on the right show the corresponding relative optical density (mean ± s.e.m., *n* = 4 independent experiments; **, *p* < 0.05, ANOVA on Ranks; ***, *p* < 0.001, ANOVA). Detailed information about the Western blotting can be found at Appendix A. (**F**) Confocal fluorescence images obtained by immunostaining with antibodies selective for Cx43 (top left, green), Cx26 (bottom left, green) and MelanA (right, red) in representative sections of melanomas grown in DSCs; nuclei were stained with DAPI; scale bar: 10 μm.

**Figure 3 cancers-13-05062-f003:**
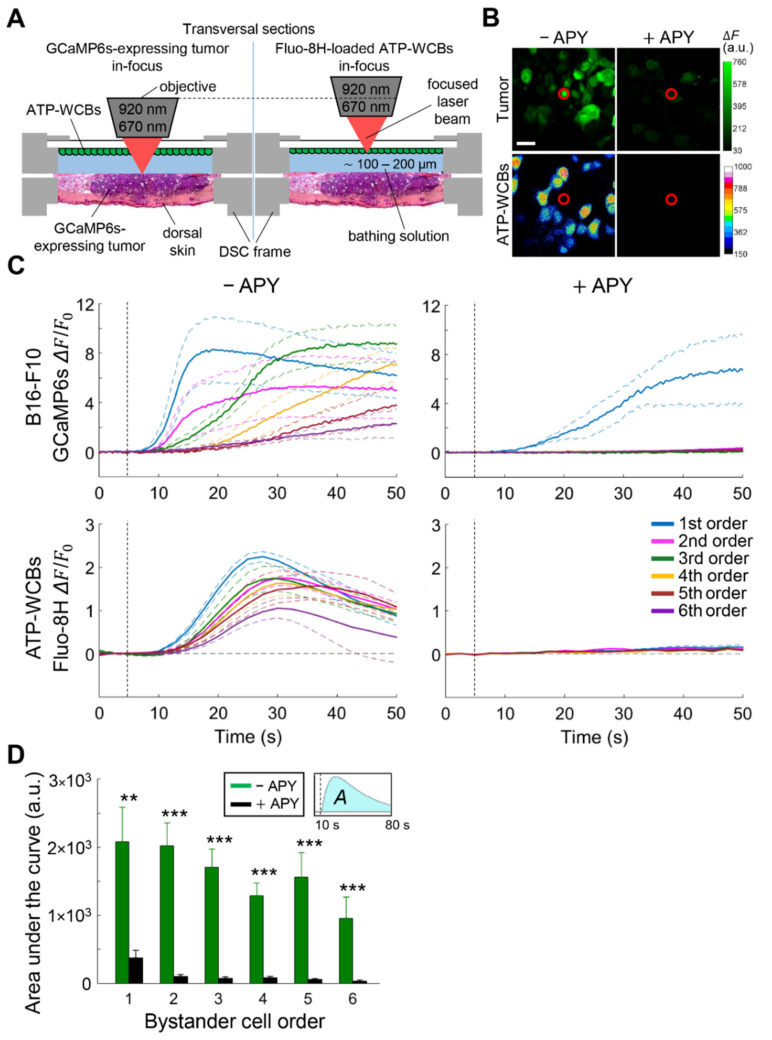
Whole-cell biosensors for adenosine triphosphate (ATP) detection (ATP-WCBs) are activated by extracellular ATP released during the propagation of calcium (Ca^2+^) waves induced by focal photodynamic therapy (fPDT) in the dorsal skinfold chamber (DSC). (**A**) Schematic representation of the multiphoton microscope objective lens oscillating between two focal planes for real-time detection of ATP release during fPDT stimulation (left, GCaMP6s-expressing tumor; right, Fluo-8H-loaded ATP-WCBs); (**B**) Representative back-projections of Δ*F* = *F* − *F*_0_ (with *F*_0_ pre-stimulus value) frames acquired from tumor (top) and ATP-WCBs (bottom) during fPDT stimulation in the absence of apyrase (−APY, left) or in its presence (+APY, 250 U/mL, right) in Ca^2+^-free extracellular medium (CFEM). Red circles mark the location of the photoactivation laser beam; scale bar: 20 µm; (**C**) Average Ca^2+^ responses of bystander melanoma cells (top) and ATP-WCBs (bottom) to fPDT before (left) and after (right) addition of APY to the extracellular medium; Δ*F*/*F*_0_ signals [mean (solid lines) ± standard error of the mean (s.e.m., dashed lines)] are shown for each bystander cell order. Results are representatives of *n* ≥ 3 experiments performed in 2 tumors. Vertical dashed lines mark the onset of laser irradiation (*t* = 5 s); (**D**) Effect of APY on the amplitudes of fPDT-induced Ca^2+^ waves in CFEM supplemented with 5 mM of EGTA; the histogram shows the area under GCaMP6s Δ*F*/*F*_0_ signals (*A*, see inset) computed for each bystander cell order in the absence of APY (−APY, green bars, also shown in Figure 1C and Figure 2A) or in its presence (+APY, black bars); data (mean ± s.e.m.) were pooled from *n* ≥ 6 experiments in at least 2 different melanomas for each condition; a.u., arbitrary units; **, *p* < 0.01; ***, *p* < 0.001; the Mann-Whitney *U* test.

**Figure 4 cancers-13-05062-f004:**
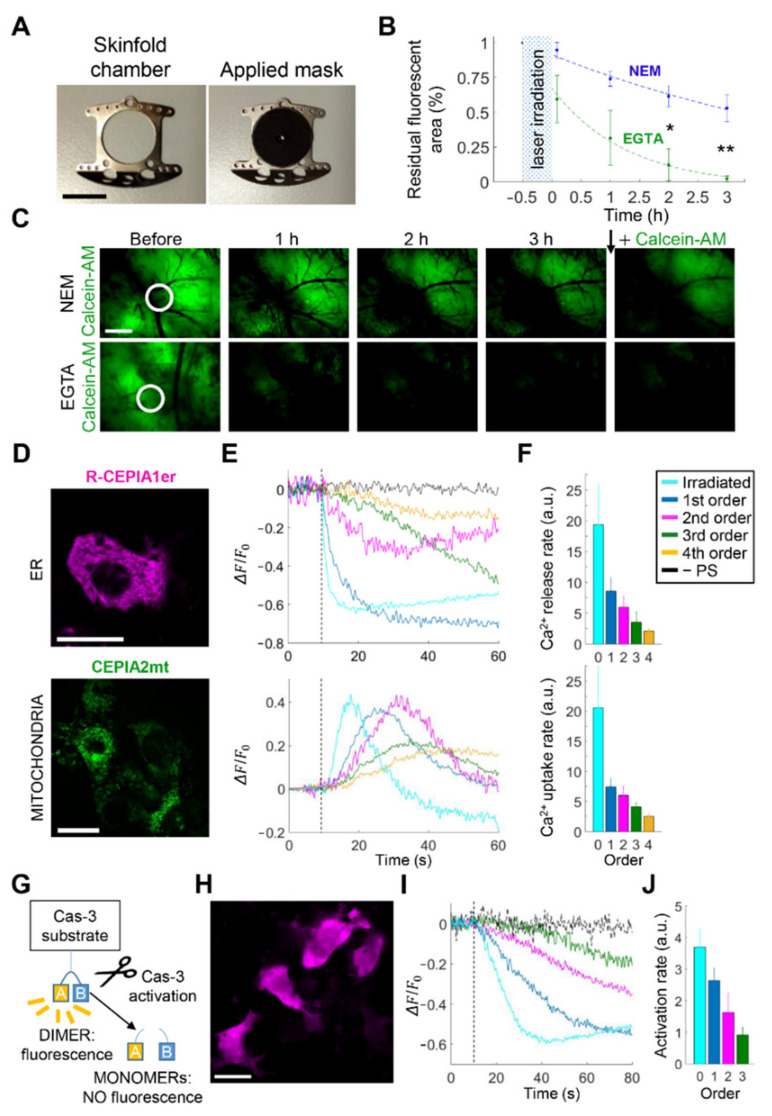
In vivo intratumor injection of ethylene glycol-bis(β-aminoethyl ether)-N,N,N′,N′-tetraacetic acid (EGTA) prior to spatially confined photodynamic therapy (scPDT) boosts bystander cell killing via calcium (Ca^2+^)-dependent apoptotic pathways. (**A**–**C**) B16-F10 melanomas grown in the dorsal skinfold chamber (DSC) were loaded with the photosensitizer (PS) and partially irradiated (scPDT, 30 min duration, irradiance ~378 mW/cm^2^) after microinjection of normal extracellular medium (NEM, control conditions) or Ca^2+^-free extracellular medium supplemented with 5 mM of EGTA (EGTA conditions). Tumor cell demise was assayed by time-lapse microscopy using calcein-AM (co-loaded with PS); (**A**) DSC titanium frame (left) with applied opaque mask (right) used to restrict melanoma irradiation in scPDT experiments (central hole diameter = 1.3 mm); scale bar: 1 cm; (**B**) Post-irradiation variation of melanoma surface area with persistent calcein-AM fluorescence in control (NEM; *n* = 4, blue) and EGTA (*n* = 3, green) conditions. Interpolating curves (dashed lines) were computed by data fitting with the function f(*t*) = 1 − *a* + *ae*^−*b*(*t*+*c*)^ (for parameter values, see Appendix A). *, *p* < 0.05; **, *p* < 0.01, two-tailed *t*-test; (**C**) Representative results of time-lapse fluorescence imaging with calcein-AM. Images were acquired before and after scPDT (within white circles) at shown time points; the black down arrow marks the time point of calcein-AM reloading in the tumors (3 h 30 min); scale bars: 1 mm; (**D**–**J**) Experiments were performed in B16-F10 syngeneic melanomas exposed to NEM, expressing one of the following genetically encoded fluorescent indicators: R-CEPIA1er, a Ca^2+^ indicator targeted to the endoplasmic reticulum (ER, **D**–**F**, top); CEPIA2mt, a Ca^2+^ indicator targeted to mitochondria (**D**–**F**, bottom); an indicator for caspase-3 (Cas-3) activation (**G**–**J**); (**D**,**H**) Representative fluorescence images of melanoma cells expressing the aforementioned Ca^2+^ (**D**) or Cas-3 indicators (**H**); scale bars: 20 μm; (**E**,**I**) Representative color-coded Δ*F*/*F*_0_ signals in the irradiated cell and surrounding bystander cells; black traces are representative results obtained in the absence of PS (−PS, negative control). The vertical dashed lines mark the onset of laser irradiation (*t* = 10 s); (**F**) Rates of Ca^2+^ signals computed up to the 4th bystander cell order as the absolute value of the average slope of the post-irradiation linear descending (ER, top) or ascending (mitochondria, bottom) trace segment. Data were pooled from *n* ≥ 10 experiments in at least 2 tumors for each condition and quoted as mean values ± standard error of the mean (s.e.m.); (**G**) Schematic representation of the mechanism of action for the fluorescent Cas-3 indicator; (**J**) Cas-3 activation rate computed up to the 3rd bystander cell order as the absolute value of the average slope of the linear descending post-irradiation trace segment. Data (mean ± s.e.m.) were pooled from *n* = 4 experiments in 2 tumors.

**Figure 5 cancers-13-05062-f005:**
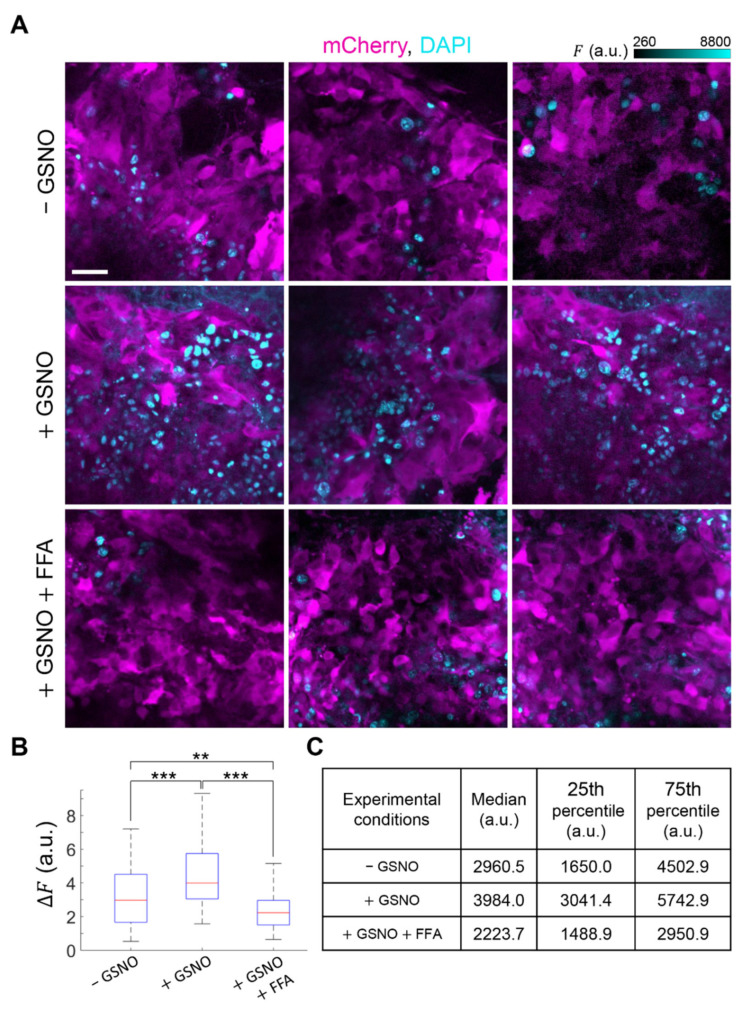
In vivo 4′,6-Diamidine-2′-phenylindole dihydrochloride (DAPI) uptake assays. (**A**) Representative fluorescence intensity (*F*) images of mCherry-expressing melanomas acquired approximately 50 µm below tumor surface after 30 min of incubation with DAPI (5 μM) dissolved in: normal extracellular medium (NEM, control conditions: −GSNO); NEM containing 1 mM of S-Nitrosoglutathione (+GSNO); NEM containing 1 mM of GSNO plus 100 µM of flufenamic acid (+GSNO +FFA); scale bar: 20 µm; a.u., arbitrary units; (**B**) Box plots showing the distributions of Δ*F* = *F*(30 min) − *F*_0_ (*F*_0_ = *F* before DAPI incubation) for DAPI fluorescence intensity measured in *n* ≥ 74 nuclei in −GSNO (3 tumors), +GSNO (3 tumors) or +GSNO +FFA (2 tumors) conditions; red horizontal bars represent the median; bottom and top edges of the blue boxes indicate the 25th and 75th percentiles, respectively; black tails of the boxes mark the most extreme data points in the distribution. **, *p* < 0.01; ***, *p* < 0.001, the Mann-Whitney *U* test; (**C**) Numerical values of median and quartiles for the three data distributions.

**Figure 6 cancers-13-05062-f006:**
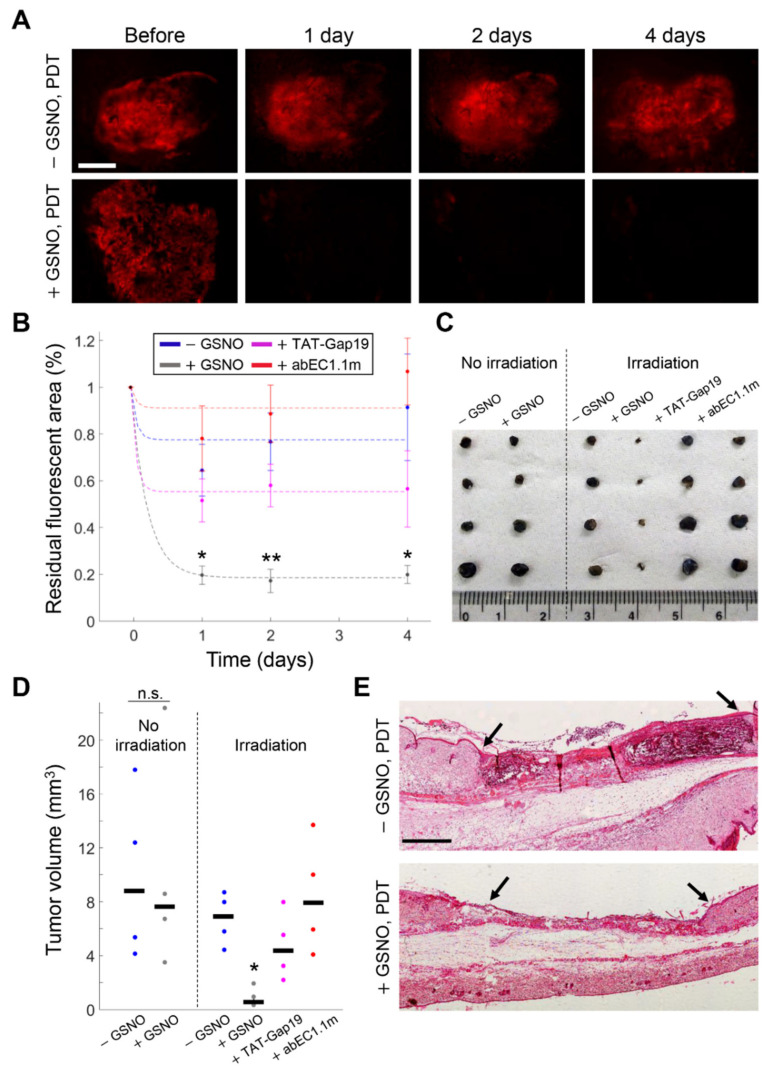
Microinjected S-Nitrosoglutathione (GSNO) enhances the efficacy of full photodynamic therapy (PDT) in murine melanoma by activating connexin hemichannels. (**A**) Photosensitizer (PS)-loaded melanomas grown in the dorsal skinfold chamber (DSC) and expressing mCherry were irradiated (full PDT, 60 min, irradiance ~245 mW/cm^2^) 20 min after microinjection of: normal extracellular medium (NEM, control conditions: −GSNO); GSNO (1 mM) dissolved in NEM (+GSNO); GSNO (1 mM) plus TAT-Gap19 (150 µM, +TAT-Gap19) dissolved in NEM or abEC1.1m (2 µM, +abEC1.1m) dissolved in NEM; tumor cell demise was assayed by time-lapse microscopy of red fluorescent mCherry. Shown are representative images acquired before and after full PDT in −GSNO and +GSNO conditions at shown time points; scale bar: 1 mm; (**B**) Melanoma surface area with persistent mCherry post-irradiation fluorescence in −GSNO (blue, *n* = 4), +GSNO (gray, *n* = 4), +TAT-Gap19 (magenta, *n* = 4) and +abEC1.1m (red, *n* = 4) conditions. Interpolating curves (dashed lines) are data fitting with the function f(*t*) = 1 − *a* + *ae*^−*b*(*t*+*c*)^ (for parameter values see Appendix A). *, *p* < 0.05; **, *p* < 0.01, the Mann-Whitney *U* test; (**C**–**E**) Melanomas were harvested 4 days after treatment. Controls were loaded with PS and microinjected but were not irradiated (No irradiation); (**C**) Tumor masses excised after treatment in each of the six conditions, scale unit: cm; (**D**) Tumor volumes measured by micro-computed tomography analysis; black horizontal bars mark the median. n.s., not significant; *, *p* < 0.05, the Kruskal-Wallis test (for post hoc pairwise comparisons see Appendix A); (**E**) Hematoxylin and eosin staining of tumor-bearing DSC tissues exposed to full PDT in −GSNO or +GSNO conditions. Black arrows mark the edges of punch biopsies where B16-F10 melanoma cells were seeded; scale bar: 1 mm.

**Figure 7 cancers-13-05062-f007:**
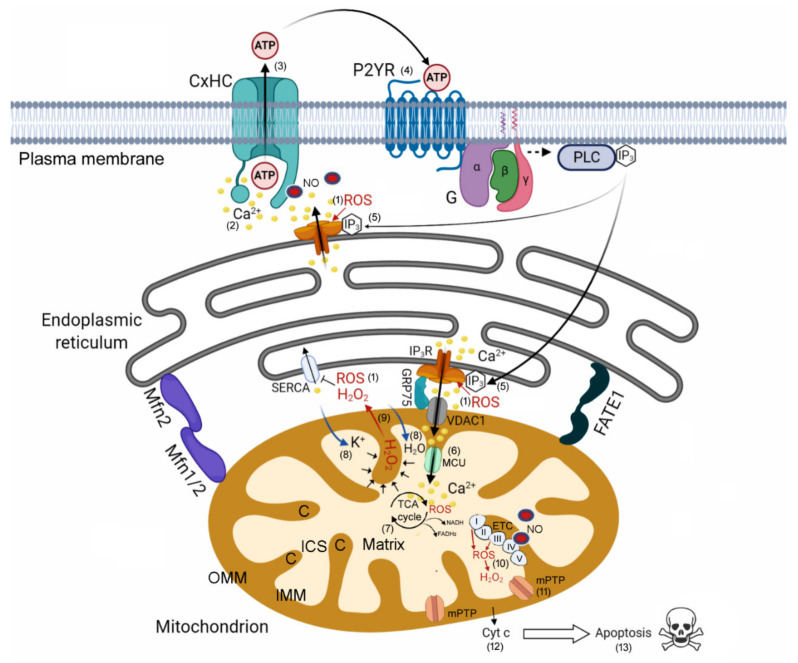
Schematic model for connexin (Cx) hemichannels (HC)-related signaling downstream of photosensitizer (PS) activation. (1) Reactive oxygen species (ROS), generated directly or indirectly after PS photoactivation [22], activate inositol 1,4,5-trisphosphate (IP_3_) receptors (Rs), promoting calcium (Ca^2+^) release from the endoplasmic reticulum (ER). (2) The rise in cytosolic Ca^2+^ concentration gates HCs from the inside, (3) permitting the release of adenosine triphosphate (ATP) from cytosol to extracellular milieu. (4) The released ATP activates metabotropic P2YRs and consequent IP_3_ production via G-protein-coupled activation of phospholipase C (PLC). (5) IP_3_ binding to IP_3_R in the ER potentiates ROS-mediated Ca^2+^ release. (6) This enhances Ca^2+^ uptake into the mitochondrial matrix via voltage-dependent anion-selective channel 1 (VDAC1) in the outer mitochondrial membrane (OMM) and mitochondrial Ca^2+^ uniporter (MCU) in the inner mitochondrial membrane (IMM). The tight spacing between ER and mitochondria, which is key for efficient ER-mitochondria Ca^2+^ transfer, is regulated by mitofusin (Mfn) proteins and fetal and adult testis-expressed 1 (FATE1). (7) Within the matrix, Ca^2+^ regulates the tricarboxylic acid (TCA) cycle by controlling the activity of three dehydrogenases, promoting increased synthesis of nicotinamide adenine dinucleotide (NADH) and flavin adenine dinucleotide (FADH_2_) and consequent augmented ATP production by complex V (ATP synthase) of the electron transport chain (ETC). (8) Ca^2+^ entry also drives potassium (K^+^) and water (H_2_O) influx into the matrix. (9) The increased internal pressure squeezes hydrogen peroxide (H_2_O_2_) out of mitochondrial cristae (C), reinforcing ROS-mediated opening of the IP_3_R and blockade of the Sarco-Endoplasmic Reticulum Ca^2+^ ATPase (SERCA) pump, which leads to irreversible ER emptying. (10) Ca^2+^ overload of the mitochondrial matrix enhances ROS levels and H_2_O_2_ mainly via complex I and III in the ETC, promoting mitochondrial permeability transition pore (mPTP) opening. (11) Prolonged opening of the mPTP causes depolarization of the IMM and swelling of the mitochondrial matrix, which ensues in the rupture of the OMM. (12) Consequently, cytochrome c (Cyt c) is released, thus promoting apoptotic cell death (13). ICS, intracristal space (see [57,58,60,61,62,63,64,65,66,67,68]). These events are exacerbated by nitric oxide (NO), a key diffusible byproduct of aluminum phthalocyanine chloride photoactivation in irradiated cells [20]. In bystander cells, NO concentration is increased above diffusion levels by Ca^2+^-dependent enzymatic production [20,21,22] and can be further increased by administration of a NO donor (S-Nitrosoglutathione), as accomplished in this article. NO favors the opening of HCs in different cell types and Cx species [48,49,69]. In this context, it potentiates the ATP release that subtends Ca^2+^ wave propagation. In addition, NO can inhibit the ETC, particularly complex IV, but also complex I, III, and II, by imparting modifications, such as S-nitrosation and nitration to selected residues [70]. Inhibition of complex IV by NO enhances the production of mitochondrial ROS [68]. The combination of NO with superoxide anion can generate peroxinitrite, a potentially harmful radical that drives nitration and oxidation of biomolecules [70].

## Data Availability

The data that support the findings of this study are available from the corresponding author upon request. Source data are provided with this paper.

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
