# Peer review of "Connexin Hemichannel Activation by S-Nitrosoglutathione Synergizes Strongly with Photodynamic Therapy Potentiating Anti-Tumor Bystander Killing"

_cancers, 2021, doi:10.3390/cancers13205062_

Round 1
Reviewer 1 Report
The authors have made substantial efforts in revising the study presented in the manuscript, including additional experiments that helped clarify the issues highlighted by the reviewer. It is essential for the authors to include these data described in the rebuttal letter as supplemental data and integrate them into the discussion in the revision.
Author Response
Additional experiments that helped clarify the issues highlighted by the reviewer in the rebuttal letter have been included as supplemental data and integrated into the discussion in revision n.2.
Reviewer 2 Report
New edits are informative and providing a solid background to go through the research goals
Author Response
We thank the reviewer for his supportive comments.
This manuscript is a resubmission of an earlier submission. The following is a list of the peer review reports and author responses from that submission.
Round 1
Reviewer 1 Report
In this study, the authors used B16-F10 melanoma cells growing in the dorsal skinfold chamber (DSC) to examine the signalling mechanism, particularly the role of connexion hemi-channel (CX-HC) activation, in mediating bystander tumour cell death induced by photodynamic therapy (PT). The results provide evidence to suggest that PT induces CX-HC activation and ATP release from irradiated tumour cells and extracellular ATP acts as paracrine or autocrine signalling to neighbouring bystander tumour cells and induces inter-cellular calcium waves via purinergic P2Y receptor(s) and subsequent ER calcium release, leading to mitochondrial calcium overload and caspase-mediated apoptosis. In addition, the study showed that co-application of S-nitrosoglutathione synergizes with PT enhances tumour cell death and prevent tumour growth. Overall, the manuscript is well written and the findings are interesting, particularly from a therapeutic point view. However, further efforts are required to confirm the underlying mechanisms proposed by the authors.
1. While the experiments using apyrase strongly suggest the role of ATP release from tumour cells in PT-induced inter-cellular calcium waves, evidence is lacking that demonstrate the expression of the purinergic receptors, particularly calcium signal-induced P2Y receptor(s) in B16-F10 melanoma cells, and show their engagement in the proposed inter-cellular calcium signalling.
2. The ER calcium is a limited source for elevating cytosolic calcium, compared to the calcium influx from extracellular space. The notion that intracellular calcium released from ER and subsequent accumulation in mitochondria can trigger mitochondrial ROS generation and MPTP opening leading to capase-3 activation and apoptotic cell death in bystander cells remains to be substantiated. The authors need to provide evidence from additional studies, for example, in vitro studies using B16-F10 melanoma cells, to show that activation of CX-HCs can result in ATP release and such endogenously release ATP or application of exogenous ATP indeed can trigger P2Y activation and IP3R-mediated ER calcium release and subsequent mitochondrial accumulation via VDAC1 and MCU. Furthermore, such mitochondrial calcium signal is sufficient to prompt mitochondrial ROS generation and MPTP opening that drives caspase-3-dependent apoptosis.
3. Fig.4B: The data shown in were from n = 2 repeats for the NEM group? Where were the SEM errors from? The authors even claim a statistically significant difference from the EGTA group?
4. Fig.4D-F: the authors should characterize the temporal relationship of the ER and mitochondrial calcium signals. Does the ER calcium release occur before, or parallel with as described by the authors (line 540), the calcium accumulation in mitochondria? In addition, theses calcium signals and caspase-3 activation appeared to decline quickly with the distance or order. The authors should comment on such an observation.
5. The schematic summary in Fig.7 is very confusing. The authors should present or distinguish what happens in the tumour cells directly exposed to irradiation and what happens in neighbouring bystander tumour cells without irradiation exposure.
Minor points:
- A careful proofread is required to remove grammar and format errors.
- Line 54: the phrase “tetraspan membrane” is vague, and please revise.
- Line 200: what does the phrase “communication-incompetent” mean?
- Remove “m =” from the text and figure legend, which creates unnecessary confusion with “n”, which refers to the number of independent repeats.
- Give accurate details of penicillin and streptomycin concentrations.
- The authors have described the equation to determine the minimal number of each experiments but they seems not to follow this, and please state clearly the minimal number of repeats they used for each experiments accordingly.
- All abbreviations should defined in their first appearance and avoid defining repeatedly.
- It is more logic to show representative data in Fig.6D and then mean data in Fig.6C, as for other experiments presented in this and other figures.
- Carefully edit the references for consistent use of the calcium symbol, and journal abbreviation (ref-61).
Author Response
- While the experiments using apyrase strongly suggest the role of ATP release from tumour cells in PT-induced inter-cellular calcium waves, evidence is lacking that demonstrate the expression of the purinergic receptors, particularly calcium signal-induced P2Y receptor(s) in B16-F10 melanoma cells, and show their engagement in the proposed inter-cellular calcium signalling.
To demonstrate the expression of purinergic receptors in B16-F10 melanoma cells, we performed immunostaining of P2Y1 and P2Y2 receptors in B16-F10-derived tumor tissue. We detected both receptors and we added these new results as a new Supplementary Figure S4.
In the main text, we added the following paragraph:
“When extracellular Ca2+ is chelated with EGTA, the subtype of P2 purinoreceptors that can still generate Ca2+ signals in response to extracellular ATP, by mobilizing Ca2+ from the ER, are the G-protein coupled P2YRs [1]. To confirm their expression in our tumuor samples, we performed immunofluorescence experiments with specific and knock-out-validated antibodies against P2Y1R and P2Y2R and determined that both were dectable (Supplementary Fig. S4).” (lines 480-485).
- The ER calcium is a limited source for elevating cytosolic calcium, compared to the calcium influx from extracellular space. The notion that intracellular calcium released from ER and subsequent accumulation in mitochondria can trigger mitochondrial ROS generation and MPTP opening leading to capase-3 activation and apoptotic cell death in bystander cells remains to be substantiated. The authors need to provide evidence from additional studies, for example, in vitro studies using B16-F10 melanoma cells, to show that activation of CX-HCs can result in ATP release and such endogenously release ATP or application of exogenous ATP indeed can trigger P2Y activation and IP3R-mediated ER calcium release and subsequent mitochondrial accumulation via VDAC1 and MCU. Furthermore, such mitochondrial calcium signal is sufficient to prompt mitochondrial ROS generation and MPTP opening that drives caspase-3-dependent apoptosis.
Compared to the calcium (Ca2+) influx from extracellular space, ER Ca2+ is certainly a limited source for elevating cytosolic Ca2+. However, Ca2+ transfer between ER and mitochondria, which occurs via VDAC1 and MCU, is critical for cellular physiology and functions, including mitochondrial metabolism and cell death (see e.g. the recent reviews in refs. [2] and [3] at the end of this document).
In the present work, we first demonstrated connexin hemichannels (Cx HCs) activity in melanoma cells in vivo by the dye uptake technique (main text, Figure 2B-D). Then, by using ATP whole-cell biosensors, we demostrated that ATP was released following focal PS photoactivation in vivo when the open probability of Cx HCs was increased by lowering the extracellular Ca2+ concentration (main text, Figure 3A-C). We also verified that the propagation of bystander effects was equally impaired in the presence of different Cx HC blockers (main text, Figure 2A) and in the absence of extracellular ATP (main text, Figure 3D). Therefore, we conclude that Cx HC activation promoted ATP release, which is a key component of the bystander effect triggered by PS photoactivation.
To address the reviewer’s question (whether application of exogenous ATP indeed can trigger P2YR activation and IP3R-mediated ER calcium release), we performed new in vitro experiments with cultured B16-F10 cells expressing either ER or mitochondrial fluorescent Ca2+-indicators (G-CEPIA1er or CEPIAmt2, respectively), or a biosensor for caspase-3 (Cas-3) activity. We stimulated the cells with pressure-application of ATP (100 nM) in Ca2+-free extracellular medium (CFEM, see Materials and Methods for composition) using a glass micropipette connected to a pneumatic Pico Pump and recorded Ca2+ signals in the organelles.
Figure A (Viewable in PDF reply) Effect of extracellular ATP on ER Ca2+ release, mitochondrial Ca2+ uptake and Cas-3 activation. ATP was applied by pressure through a glass micro-capillary in the absence of extracellular Ca2+. (a) Application of ATP (100 nM, red traces) in CFEM promotes Ca2+ release from the ER. Shown are average signals (solid lines) ± s.e.m. (dashed lines) obtained from n ≥ 5 G-CEPIA1er-expressing cells in both conditions. The black traces are the results of negative control experiments in which ATP was omitted from the pressure-applied medium. (b) Application of ATP in CFEM (100 nM, red traces) triggers mitochondrial Ca2+ uptake. Also in this case, the black traces are the results of negative control experiments in which ATP was omitted from the pressure-applied medium. Shown are average signals (solid lines) ± s.e.m. (dashed lines) obtained from n ≥ 10 CEPIAmt2-expressing cells in both conditions. (c) Application of ATP (100 nM) in CFEM triggers Cas-3 activation only cell cultures that had been pre-incubated for 50 min with GSNO (100 nM, blue traces). Black traces are the result of ATP application experiments in the absence of GSNO. Shown are average signals (solid lines) ± s.e.m. (dashed lines) obtained from n ≥ 7 cells expressing the fluorescent indicator for Cas-3 activity in both conditions. Vertical dashed lines mark the onset and the end of stimulus delivery. At the bottom of each graph, point-by-point p-values (two-sample t-test) between the curves obtained in the two conditions are shown on a log scale; horizontal blue lines mark the level of statistical significance (P = 0.05).
As shown in the new Figure A, in the absence of extracellular Ca2+, ATP promoted Ca2+ release from the ER (panel (a)) and Ca2+ uptake in mitochondria (panel (b)). However, ATP triggered Cas-3 activation only in cell cultures that had been pre-incubated with 100 µM S-Nitrosoglutathione (GSNO, panel (c)). This is consistent with the rest of our results and highligts the importance of nitric oxide (NO) for the induction of cell death and tumor eradication in these experimental conditions. See also reply to question 5, below.
- Fig.4B: The data shown in were from n = 2 repeats for the NEM group? Where were the SEM errors from? The authors even claim a statistically significant difference from the EGTA group?
We added the results of two more trials to the NEM group in the Calcein-AM assay, recomputed statistical significance and replaced the graph in Figure 4B and the exponential fitting parameters in Supplementary Table S3.
- Fig.4D-F: the authors should characterize the temporal relationship of the ER and mitochondrial calcium signals. Does the ER calcium release occur before, or parallel with as described by the authors (line 540), the calcium accumulation in mitochondria? In addition, theses calcium signals and caspase-3 activation appeared to decline quickly with the distance or order. The authors should comment on such an observation.
Figure B.(Viewable in PDF reply) Rates of Ca2+ signals in the ER (R-CEPIA1er, black bars) and mitochondria (CEPIA2mt, white bars) of the irradiated cell and bystander cells following in vivo fPDT. Data were pooled from n ≥ 10 experiments in at least 2 tumors for each Ca2+ indicator and quoted as mean values ± s.e.m.. p-values (P) were computed by the Mann-Whitney U test.
To characterize the temporal relationship between the optical signals obtained in vivo from R-CEPIA1er (Ca2+ in the ER, Kd = 565 mM, Fmax/Fmin = 8.8) and CEPIA2mt (mitochondrial Ca2+, Kd = 160 nM, Fmax/Fmin = 1.7), we generated the data in Figure B by first normalizing each trace from either biosensor to 1. This allows, at least, to compensate for differences in the dynamic ranges of the two indicators. Thereafter, we computed the time derivative of these normalized signals and averaged the results within a time interval of 10 s following PS photoactivation (in the main text, the graphs of Figure 4F and 4L were replaced with the results of the new calculations). Statistical comparison showed no significant differences in the rates of the two signals, neither in the irradiated nor in bystander cells.
To interpret these data, it is necessary to consider that mitochondria uptake Ca2+ from the cytosol where, in our conditions, Ca2+ is released from the ER. However, this does not necessarily translate into a delayed optical signal from the mitochondria relative to the ER, due to factors such as binding kinetics and concentration of the Ca2+-sensitive fluorescent indicators. In addition, Ca2+ in the lumen of the ER is heavily buffered. Thus, relatively large amounts of Ca2+ can be released from the ER despite small variations of the free Ca2+ concentration in the lumen, which is what the ER Ca2+ indicator responds to. Indeed, as shown in Figure B above, the processes of Ca2+ release from the ER and Ca2+ uptake in mitochondria appear to occur simultaneously in our imaging experiments. For these reasons, we causciously used the sentence “Following focal PS activation in NEM, these indicators revealed a rapid release of Ca2+ from the ER (Fig. 4E and F, top) paralleled by Ca2+ uptake into mitochondria (Fig. 4E and F, bottom) in both irradiated and bystander cells” (lines 558-560).
As for the decline of Ca2+ signals and Cas-3 activation with the bystander cell order, it is worth considering that data from the fluorescent Ca2+-sensitive indicators were acquired within the first 80 seconds after photoactivating the PS in a single cell within the tumor mass. However, the processes induced by PS photoactivation in bystander cells continue well after the time window of our imaging experiments in a domino-like effect typical of critical phenomena. Indeed, we tracked the fate of bystander cells for hours and days after PS photoactivation and revealed massive cell loss, and even dramatic reduction of tumor volume after full PDT in GSNO conditions, as reported in the Results.
- The schematic summary in Fig.7 is very confusing. The authors should present or distinguish what happens in the tumour cells directly exposed to irradiation and what happens in neighbouring bystander tumour cells without irradiation exposure.
The scheme refers to events that take place both in irradiated and in bystander cells downstream of PS activation. However, to avoid confusion, we eliminated the PS from the drawing. We also made other minor changes to better identify mitochondrial structures, and added NO as a gating mechanism for CX HCs (refs. [4-6]) as well as an inhibitor of the mitochondrial electron transport chain (refs. [3,7]).
Figure 7 (revised, viewable in PDF reply) Schematic model for HC-related signaling downstream of PS activation. (1) ROS, generated directly or indirectly after PS photoactivation (Ref. [8]), activate IP3Rs promoting Ca2+ release from the ER. (2) The rise of [Ca2+]cyt gates HCs from the inside, (3) permitting the release of ATP from cytosol to extracellular milieu. (4) The released ATP activates metabotropic P2YRs and consequent IP3 production via G-protein-coupled activation of phospholipase C (PLC). (5) IP3 binding to IP3R in the ER potentiates ROS-mediated Ca2+ release. (6) This enhances Ca2+ uptake into the mitochondrial matrix via voltage-dependent anion-selective channel 1 (VDAC1) in the outer mitochondrial membrane (OMM) and mitochondrial Ca2+ uniporter (MCU) in the inner mitochondrial membrane (IMM). The tight spacing between ER and mitochondria, that is key for efficient ER-mitochondria Ca2+ transfer, is regulated by mitofusin (Mfn) proteins and Fetal and Adult Testis-Expressed 1 (FATE1). (7) Within the matrix, Ca2+ regulates the tricarboxylic acid (TCA) cycle by controlling the activity of three dehydrogenases, promoting increased synthesis of NADH and FADH2 and consequent augmented ATP production by complex V (ATP synthase) of the electron transport chain (ETC). (8) Ca2+ entry also drives K+ and water (H2O) influx into the matrix. (9) The increased internal pressure squeezes H2O2 out of mitochondrial cristae (C), reinforcing ROS-mediated opening of the IP3R and blockade of the SERCA pump, which leads to irreversible ER emptying. (10) Ca2+ overload of the mitochondrial matrix enhances ROS level and H2O2 mainly via Complex I and III in the ETC, promoting mitochondrial permeability transition pore (mPTP) opening. (11) Prolonged opening of the MPTP causes depolarization of the IMM and swelling of the mitochondrial matrix, which ensues in the rupture of the OMM. (12) Consequently, cytochrome c (Cyt c) is released, thus promoting apoptotic cell death (13). C, Cristae; ICS, intracristal space (see refs. [2,3,9-17]). These events are exacerbated by NO, a key diffusible byproduct of AlPcCl photoactivation in irradiated cells (ref. [18]). In bystander cells, NO concentration is increased above diffusion levels by Ca2+-dependent enzymatic production (refs.[8,18,19]), and can be further increased by administration of a NO donor (GSNO) as done in this article. NO favors the opening of HCs in different cell type and Cx species (Refs. [4-6]). In this context, it potentiates the ATP release that subtends Ca2+ wave propagation. In addition, NO can inhibit the ETC, particularly Complex IV, but also Complex I, III and II, by imparting modifications such as S-nitrosation and nitration to selected residues (Ref. [7]). Inhibition of Complex IV by NO enhances the production of mitochondrial ROS (Ref. [3]). The combination of NO with O2¯¯âˆ™ can generate peroxinitrite, a potentially harmful radical that drives nitration and oxidation of biomolecules (Ref. [7]).
Minor points:
1. A careful proofread is required to remove grammar and format errors.
Done.
2. Line 54: the phrase “tetraspan membrane” is vague, and please revise.
We replaced “tetraspan membrane” with the phrase “membrane proteins with four transmembrane helices” (line 56).
3. Line 200: what does the phrase “communication-incompetent” mean?
For clarity, we added to the text: “(i.e. lacking ICGJ-mediated communication)” (lines 191-192).
4. Remove “m =” from the text and figure legend, which creates unnecessary confusion with “n”, which refers to the number of independent repeats.
We accepted the Reviewer’s suggestion and revised the figure legends and the text.
5. Give accurate details of penicillin and streptomycin concentrations.
Done.
6. The authors have described the equation to determine the minimal number of each experiments but they seems not to follow this, and please state clearly the minimal number of repeats they used for each experiments accordingly.
We better clarified this point by revising the Materials and Methods section.
7. All abbreviations should defined in their first appearance and avoid defining repeatedly.
We revised inappropriate definitions of abbreviations throughout all text and we added the list of abbreviations at the end of the manuscript for the reader convenience.
8. It is more logic to show representative data in Fig.6D and then mean data in Fig.6C, as for other experiments presented in this and other figures.
We accepted the Reviewer’s suggestion and corrected Figure 6.
9. Carefully edit the references for consistent use of the calcium symbol, and journal abbreviation (ref-61).
The references were automatically generated using an Endnote. We preferred to stick to the bibliography software package, as recommended in the Instructions for Authors given by the journal Cancers.
References
- Woods, L.T.; Forti, K.M.; Shanbhag, V.C.; Camden, J.M.; Weisman, G.A. P2Y receptors for extracellular nucleotides: Contributions to cancer progression and therapeutic implications. Biochemical pharmacology 2021, 187, 114406, doi:10.1016/j.bcp.2021.114406.
- Loncke, J.; Kaasik, A.; Bezprozvanny, I.; Parys, J.B.; Kerkhofs, M.; Bultynck, G. Balancing ER-Mitochondrial Ca(2+) Fluxes in Health and Disease. Trends in cell biology 2021, 31, 598-612, doi:10.1016/j.tcb.2021.02.003.
- Modesti, L.; Danese, A.; Angela Maria Vitto, V.; Ramaccini, D.; Aguiari, G.; Gafa, R.; Lanza, G.; Giorgi, C.; Pinton, P. Mitochondrial Ca(2+) Signaling in Health, Disease and Therapy. Cells 2021, 10, doi:10.3390/cells10061317.
- Retamal, M.A.; Cortes, C.J.; Reuss, L.; Bennett, M.V.; Saez, J.C. S-nitrosylation and permeation through connexin 43 hemichannels in astrocytes: induction by oxidant stress and reversal by reducing agents. Proceedings of the National Academy of Sciences of the United States of America 2006, 103, 4475-4480, doi:10.1073/pnas.0511118103.
- García, I.E.; Sánchez, H.A.; Martínez, A.D.; Retamal, M.A. Redox-mediated regulation of connexin proteins; focus on nitric oxide. Biochimica et biophysica acta. Biomembranes 2018, 1860, 91-95, doi:10.1016/j.bbamem.2017.10.006.
- Retamal, M.A.; Schalper, K.A.; Shoji, K.F.; Bennett, M.V.; Saez, J.C. Opening of connexin 43 hemichannels is increased by lowering intracellular redox potential. Proceedings of the National Academy of Sciences of the United States of America 2007, 104, 8322-8327, doi:10.1073/pnas.0702456104.
- Tengan, C.H.; Moraes, C.T. NO control of mitochondrial function in normal and transformed cells. Biochim Biophys Acta Bioenerg 2017, 1858, 573-581, doi:10.1016/j.bbabio.2017.02.009.
- Nardin, C.; Peres, C.; Mazzarda, F.; Ziraldo, G.; Salvatore, A.M.; Mammano, F. Photosensitizer Activation Drives Apoptosis by Interorganellar Ca(2+) Transfer and Superoxide Production in Bystander Cancer Cells. Cells 2019, 8, doi:10.3390/cells8101175.
- Rizzuto, R.; Pinton, P.; Carrington, W.; Fay, F.S.; Fogarty, K.E.; Lifshitz, L.M.; Tuft, R.A.; Pozzan, T. Close contacts with the endoplasmic reticulum as determinants of mitochondrial Ca2+ responses. Science 1998, 280, 1763-1766, doi:10.1126/science.280.5370.1763.
- Periasamy, M.; Kalyanasundaram, A. SERCA pump isoforms: their role in calcium transport and disease. Muscle & nerve 2007, 35, 430-442, doi:10.1002/mus.20745.
- Kaplan, P.; Babusikova, E.; Lehotsky, J.; Dobrota, D. Free radical-induced protein modification and inhibition of Ca2+-ATPase of cardiac sarcoplasmic reticulum. Mol Cell Biochem 2003, 248, 41-47, doi:10.1023/a:1024145212616.
- De Stefani, D.; Patron, M.; Rizzuto, R. Structure and function of the mitochondrial calcium uniporter complex. Biochimica et biophysica acta 2015, 1853, 2006-2011, doi:10.1016/j.bbamcr.2015.04.008.
- Booth, D.M.; Enyedi, B.; Geiszt, M.; Varnai, P.; Hajnoczky, G. Redox Nanodomains Are Induced by and Control Calcium Signaling at the ER-Mitochondrial Interface. Molecular cell 2016, 63, 240-248, doi:10.1016/j.molcel.2016.05.040.
- Kerkhofs, M.; Bittremieux, M.; Morciano, G.; Giorgi, C.; Pinton, P.; Parys, J.B.; Bultynck, G. Emerging molecular mechanisms in chemotherapy: Ca(2+) signaling at the mitochondria-associated endoplasmic reticulum membranes. Cell Death Dis 2018, 9, 334, doi:10.1038/s41419-017-0179-0.
- Joseph, S.K.; Young, M.P.; Alzayady, K.; Yule, D.I.; Ali, M.; Booth, D.M.; Hajnoczky, G. Redox regulation of type-I inositol trisphosphate receptors in intact mammalian cells. The Journal of biological chemistry 2018, 293, 17464-17476, doi:10.1074/jbc.RA118.005624.
- Rimessi, A.; Pedriali, G.; Vezzani, B.; Tarocco, A.; Marchi, S.; Wieckowski, M.R.; Giorgi, C.; Pinton, P. Interorganellar calcium signaling in the regulation of cell metabolism: A cancer perspective. Seminars in cell & developmental biology 2019, doi:10.1016/j.semcdb.2019.05.015.
- Loncke, J.; Kerkhofs, M.; Kaasik, A.; Bezprozvanny, I.; Bultynck, G. Recent advances in understanding IP3R function with focus on ER-mitochondrial Ca2+ transfers. Current Opinion in Physiology 2020, 17, 9, doi:https://doi.org/10.1016/j.cophys.2020.07.011.
- Cali, B.; Ceolin, S.; Ceriani, F.; Bortolozzi, M.; Agnellini, A.H.; Zorzi, V.; Predonzani, A.; Bronte, V.; Molon, B.; Mammano, F. Critical role of gap junction communication, calcium and nitric oxide signaling in bystander responses to focal photodynamic injury. Oncotarget 2015, 6, 10161-10174, doi:10.18632/oncotarget.3553.
- Giorgi, C.; Bonora, M.; Missiroli, S.; Poletti, F.; Ramirez, F.G.; Morciano, G.; Morganti, C.; Pandolfi, P.P.; Mammano, F.; Pinton, P. Intravital imaging reveals p53-dependent cancer cell death induced by phototherapy via calcium signaling. Oncotarget 2015, 6, 1435-1445, doi:10.18632/oncotarget.2935.

Reviewer 2 Report
The manuscript “Connexin Hemichannel Activation by S-nitrosoglutathione Synergizes Strongly With Photodynamic Therapy Potentiating Anti-Tumor Bystander Killing” by Nardin et al. demonstrate bystander effects in PDT proceed by conveying death signals from irradiated to bystander tumor cells. These findings have a clear translational significance and suggest that these signaling mechanisms can be amplified to improve tumor eradication.
The use of a clinically relevant tumor model is one of the major strengths of the study. However, However, there is no discussion of these interesting data in the text. Other issues are of major concern and need to be addressed before consideration for publication. I find the manuscript is very carefully put together with statements made with a clear rationale. It would be great to add some introduction about the bystander effects triggered by the activation of a certain class of photosensitizers or a general case with all photosensitizers, one would expect a broad overview on that.
Author Response
The use of a clinically relevant tumor model is one of the major strengths of the study. However, there is no discussion of these interesting data in the text.
We thank the Reviewer for this comment. We have incorporated the phrase “clinically relevant tumor model” in the conclusions. As for the discussion of the clinical relevance of our model, we devoted a section (paragraph 4.2 of the discussion) to highlighting the translational potential of our results.
It would be great to add some introduction about the bystander effects triggered by the activation of a certain class of photosensitizers or a general case with all photosensitizers, one would expect a broad overview on that.
We added Refs. 3 to 8 to the introduction, to expand the issue of bystander effects triggered by a variety of photosensitizers.